# ZARTS: On Zero-order Optimization for Neural Architecture Search

## Abstract

Differentiable architecture search (DARTS) has been a popular one-shot paradigm for NAS due to its high efficiency. It introduces trainable architecture parameters to represent the importance of candidate operations and proposes first/second-order approximation to estimate their gradients, making it possible to solve NAS by gradient descent algorithm. However, our in-depth empirical results show that the approximation will often distort the loss landscape, leading to the biased objective to optimize and in turn inaccurate gradient estimation for architecture parameters. This work turns to zero-order optimization and proposes a novel NAS scheme, called ZARTS, to search without enforcing the above approximation. Specifically, three representative zero-order optimization methods are introduced: RS, MGS, and GLD, among which MGS performs best by balancing the accuracy and speed. Moreover, we explore the connections between RS/MGS and gradient descent algorithm and show that our ZARTS can be seen as a robust gradient-free counterpart to DARTS. Extensive experiments on multiple datasets and search spaces show the remarkable performance of our method. In particular, results on 12 benchmarks verify the outstanding robustness of ZARTS, where the performance of DARTS collapses due to its known instability issue. Also, we search on the search space of DARTS to compare with peer methods, and our discovered architecture achieves 97.54% accuracy on CIFAR-10 and 75.7% top-1 accuracy on ImageNet, which are state-of-the-art performance.

## 1 Introduction

Despite their success, neural networks are still designed mainly by humans (Simonyan & Zisserman, 2014; He et al., 2016; Howard et al., 2017). It remains open to automatically discover effective and efficient architectures. The problem of neural architecture search (NAS) has attracted wide attention, which can be modeled as bi-level optimization for network architectures and operation weights.

One-shot NAS (Bender et al., 2018) is a popular search framework that regards neural architectures as directed acyclic graphs (DAG) and constructs a supernet with all possible connections and operations in the search space. DARTS (Liu et al., 2019) further introduces trainable architecture parameters to represent the importance of candidate operations, which are alternately trained by SGD optimizer along with network weights. It proposes a first-order approximation to estimate the gradients of architecture parameters, which is biased and may lead to the severe instability issue shown by (Bi et al., 2019). Other works (Zela et al., 2020b; Chen & Hsieh, 2020) point out that architecture parameters will converge to a sharp local minimum resulting in the instability issue and introduces extra regularization items so that architecture parameters converge to a flat local minimum.

In this paper, we empirically show that the first-order approximation of optimal network weights sharpens the loss landscape and results in the instability issue of DARTS. It also shifts the global minimum, misleading the training of architecture parameters. To this end, we discard such approximation and turn to zero-order optimization algorithms, which can run without the requirement that the search loss is differentiable w.r.t. architecture parameters. Specifically, we introduce a novel NAS scheme named ZARTS, which outperforms DARTS by a large margin and can discover efficient architectures stably on multiple public benchmarks.

In a nutshell, this paper sheds light on the frontier of NAS in the following aspects:

**1) Establishing zero-order based robust paradigm to solve bi-level optimization for NAS.** Differentiable architecture search has been a well-developed area (Liu et al., 2019; Xu et al., 2020b; Wang et al., 2020b) which solves the bi-level optimization of NAS by gradient descent algorithms. However, this paradigm suffers from the instability issue during search since biased approximation for optimal network weights distorts the loss landscape, as shown in Fig. 1 (a) and (b). To this end, we propose a flexible zero-order optimization NAS framework to solve the bi-level optimization problem, which is compatible with multiple potential gradient-free algorithms in the literature.

**2) Uncovering the connection between zero-order architecture search and DARTS.** This work introduces three representative zero-order optimization algorithms without enforcing the unverified differentiability assumption for search loss w.r.t. architecture parameters. We reveal the connections between the zero-order algorithms and gradient descent algorithm, showing that two implementations of ZARTS can be seen as gradient-free counterparts to DARTS, being more stable and robust.

**3) Strong empirical performance and robustness.** Experiments on four datasets and five search spaces have been conducted to evaluate the performance of our method. Unlike DARTS that suffers the severe instability issue shown by (Zela et al., 2020b; Bi et al., 2019), ZARTS can stably discover effective architectures on various benchmarks. In particular, the searched architecture achieves 75.7% top-1 accuracy on ImageNet, outperforming DARTS and most of its variants.

## 2 RELATED WORK

**One-shot Neural Architecture Search.** Bender et al. (2018) construct a supernet so that all candidate architectures can be seen as its sub-graph. DARTS (Liu et al., 2019) introduces architecture parameters to represent the importance of operations in the supernet and update them by gradient descent algorithm. Some works (Xu et al., 2020b; Wang et al., 2020b; Dong & Yang, 2019) reduce the memory requirement of DARTS in the search process. Other works (Zela et al., 2020b; Chen & Hsieh, 2020) point out the instability issue of DARTS, i.e., skip-connection gradually dominates the normal cells, leading to performance collapse during the search stage.

**Bi-level Optimization for NAS.** NAS can be modeled as a bi-level optimization for architecture parameters and network weights. DARTS (Liu et al., 2019) proposes first/second-order approximations to estimate gradients of architecture parameters so that they can be trained by gradient descent algorithms. However, we show that such approximation will distort the loss landscape and mislead the training of architecture parameters. Amended-DARTS (Bi et al., 2019) derives an analytic formula of the gradient w.r.t. architecture parameters that includes the inverse of Hessian matrix of network weights, which is even unfeasible to compute. In contrast, this work discards the approximation in DARTS and attempts to solve the bi-level optimization by gradient-free algorithms.

**Zero-order Optimization.** Unlike gradient-based optimization methods that require the objective differentiable w.r.t. the parameters, zero-order optimization can train parameters when the gradient of objective is unavailable or difficult to obtain, which has been widely used in adversarial robustness for neural networks (Chen et al., 2017; Ilyas et al., 2018), meta learning (Song et al., 2020), and transfer learning (Tsai et al., 2020). Liu et al. (2020b) aim at AutoML and utilize zero-order optimization to discover optimal configurations for ML pipelines. In this work, (to our best knowledge) we make the first attempt to apply zero-order optimization to NAS and experiment with multiple algorithms, from vanilla random search (Flaxman et al., 2004) to more advanced and effective direct search (Golovin et al., 2020), showing its great superiority against gradient-based methods.

## 3 BI-LEVEL OPTIMIZATION IN DARTS

Following one-shot NAS (Bender et al., 2018), DARTS constructs a supernet stacked by normal cells and reduction cells. Cells in the supernet are represented by directed acyclic graphs (DAG) with $N$ nodes $\{x_i\}_{i=1}^N$, which represents latent feature maps. Each edge $e_{i,j}$ contains multiple operations $\{o_{i,j}, o \in \mathcal{O}\}$, whose importance is represented by architecture parameters $\alpha_{i,j}^o$. Therefore, NAS can be modeled as a bi-level optimization problem by alternately updating the operation weights $\boldsymbol{\omega}$ (parameters within candidate operations on each edge) and the architecture parameters $\boldsymbol{\alpha}$:

$$\min_{\boldsymbol{\alpha}} \ \mathcal{L}_{val}(\boldsymbol{\omega}^*(\boldsymbol{\alpha}), \boldsymbol{\alpha}); \qquad \text{s.t.} \ \boldsymbol{\omega}^*(\boldsymbol{\alpha}) = \arg\min_{\boldsymbol{\omega}} \mathcal{L}_{train}(\boldsymbol{\omega}, \boldsymbol{\alpha}). \tag{1}$$

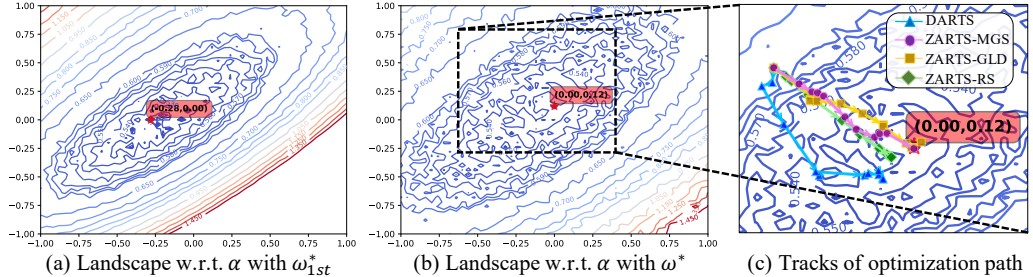

(a) Landscape w.r.t. $\alpha$ with $\omega_{1st}^*$     (b) Landscape w.r.t. $\alpha$ with $\omega^*$     (c) Tracks of optimization path

Figure 1: Loss landscapes w.r.t. architecture parameters $\alpha$ where the red star indicates the global minimum. (a) illustrates the landscape with $\omega_{1st}^*$. (b) illustrates the landscape with $\omega^*$, which is obtained by training $\omega$ for 10 iterations. To fairly compare the landscapes in (a) and (b), we utilize the same model and candidate $\alpha$ points. We observe the first-order approximation sharpens the landscape. (c) displays the tracks of optimization path of DARTS and our ZARTS. Starting at the same initial point, ZARTS can converge to the global minimum but DARTS fails.

## 3.1 FUNDAMENTAL LIMITATIONS IN THE DARTS FRAMEWORK

By enforcing an unverified (and in fact difficult to verify) assumption that the search loss $\mathcal{L}_{val}(\omega^*(\alpha), \alpha)$ is differentiable w.r.t. $\alpha$, DARTS (Liu et al., 2019) proposes a second-order approximation for the optimal weights $\omega^*(\alpha)$ by applying one-step gradient descent:

$$\omega^*(\alpha) \approx \omega_{2nd}^*(\alpha) = \omega - \xi \nabla_\omega \mathcal{L}_{train}(\omega, \alpha) = \omega', \tag{2}$$

where $\xi$ is the learning rate to update network weights. Thus the gradient of the loss function w.r.t. $\alpha$, $\nabla_\alpha \mathcal{L}_{val}(\omega^*(\alpha), \alpha)$, can be computed by the chain rule: $\nabla_\alpha \mathcal{L}_{val}(\omega^*(\alpha), \alpha) \approx \nabla_\alpha \mathcal{L}_{val}(\omega', \alpha) - \xi \nabla_{\alpha,\omega}^2 \mathcal{L}_{train}(\omega, \alpha) \nabla_{\omega'} \mathcal{L}_{val}(\omega', \alpha)$. Nevertheless, the second-order partial derivative is hard to compute, so the authors adopt the difference method, which is proved in Appendix A.1.

To further reduce the computational cost, first-order approximation is introduced by assuming $\omega^*(\alpha)$ being independent of $\alpha$, as shown in Eq. 3, which is much faster and widely used in many variants of DARTS (Chen et al., 2019; Wang et al., 2020b; Zela et al., 2020b).

$$\omega^*(\alpha) \approx \omega_{1st}^*(\alpha) = w. \tag{3}$$

The gradient is then simplified as: $\nabla_\alpha \mathcal{L}_{val}(\omega^*(\alpha), \alpha) \approx \nabla_\alpha \mathcal{L}_{val}(\omega, \alpha)$, which, however, further exacerbates the estimation bias.

Reexamining the definition of $\omega^*(\alpha)$ in Eq. 1, one would note that it is intractable to derive a mathematical expression for $\omega^*(\alpha)$, making $\mathcal{L}_{val}(\omega^*(\alpha), \alpha)$ even non-differentiable w.r.t. $\alpha$. Yet DARTS has to compromise with such approximations as Eq. 2 and Eq. 3 so that differentiability is established and SGD can be applied. However, such sketchy estimation of optimal operation weights can distort the loss landscape w.r.t. architecture parameters and thus mislead the search procedure, which is shown in Fig. 1 and analyzed in the next section.

## 3.2 DISTORTED LANDSCAPE AND INCORRECT OPTIMIZATION PROCESS IN DARTS

Fig. 1 illustrates the loss landscape with perturbations on architecture parameters $\alpha$, showing how different approximations of $\omega^*$ affect the search process. We train a supernet for 50 epochs and randomly select two orthonormal vectors as the directions to perturb $\alpha$. The same group of perturbation directions is used to draw landscapes in Fig. 1(a) and (b) for a fair comparison. Fig. 1(a) shows the loss landscape with the first-order approximation in DARTS, $\omega_{1st}^*(\alpha) = \omega$, while Fig. 1(b) shows the loss landscape with more accurate $\omega^*(\alpha)$, which is obtained by fine-tuning the network weights $\omega$ for 10 iterations for each $\alpha$. Landscapes (contours) are plotted by evaluating $\mathcal{L}$ at grid points ranged from -1 to 1 at an interval of 0.02 in both directions. Global minima are marked with stars on the landscapes, from which we have two observations: 1) The approximation $\omega_{1st}^*(\alpha) = \omega$ shifts the global minimum and sharpens the landscape [1], which is the representative characteristic of instability issue as pointed out by (Zela et al., 2020b). 2) Accurate estimation for $\omega^*$ leads to a flatter landscape, indicating that the instability issue can be alleviated. Moreover, we display the landscape with second-order approximation $\omega_{2nd}^*$ in Appendix A.2, which is also sharp but slightly flatter than

---

[1] A "sharp" landscape has denser contours than a "flat" one.

---

**Algorithm 1:** ZARTS: Zero-order Optimization Framework for Architecture Search

---

**Hyper-parameters:**

    Operation weights $\boldsymbol{\omega}$, architecture parameters $\boldsymbol{\alpha}$, sampling number $N$, iteration number $M$, update estimation function $\phi(\cdot)$.

**while** *not converged* **do**

    **Sample candidates:** $\{\mathbf{u}_i\}_{i=1}^N$, and estimate optimal operation weights $\boldsymbol{\omega}^*(\boldsymbol{\alpha}_i^\pm)$ by descending $\nabla_{\boldsymbol{\omega}} \mathcal{L}_{train}(\boldsymbol{\omega}, \boldsymbol{\alpha}_i^\pm)$ for $M$ iterations, where $\boldsymbol{\alpha}_i^\pm = \boldsymbol{\alpha} \pm \mathbf{u}_i$;

    **Compute descent direction:** $\mathbf{u}^* = \phi\left(\{\mathbf{u}_i, \boldsymbol{\omega}^*(\boldsymbol{\alpha}_i^\pm)\}_{i=1}^N\right)$;

    **Update architecture parameters:** $\boldsymbol{\alpha} \leftarrow \boldsymbol{\alpha} + \mathbf{u}^*$;

$^\star$ The sampling strategies and update estimation functions $\phi(\cdot)$ for three different zero-order optimization algorithms are detailed in Table 1.

---

Table 1: Configuration of three methods used in the ZARTS scheme. The main difference lies in the meaning of function $\phi(\cdot)$: RS follows the traditional gradient estimation algorithms, MGS estimates the update according to the improvement of loss function, while GLD uses direct search. Note that the ZARTS framework is general and can support more configurations besides the listed ones.

| Algorithm | Sampling strategy | Update estimation function $\phi\left(\{\mathbf{u}_i, \boldsymbol{\omega}^*(\boldsymbol{\alpha}_i)\}_{i=1}^N\right)$ |
|---|---|---|
| ZARTS-RS | $\mathbf{u}_i \sim q(\mathbf{u}|\boldsymbol{\alpha})$, any spherically symmetric distribution. | $\mathbf{u}^* = -\xi \cdot \frac{\varphi(d)}{2\mu N} \sum_{i=1}^N \left[\mathcal{L}(\boldsymbol{\alpha} + \mu\mathbf{u}_i) - \mathcal{L}(\boldsymbol{\alpha} - \mu\mathbf{u}_i)\right]\mathbf{u}_i$ (Eq. 6) |
| ZARTS-MGS | $\mathbf{u}_i \sim q(\mathbf{u}|\boldsymbol{\alpha})$, any proposal distribution. | $\mathbf{u}^* = \sum_{i=1}^N \left[\frac{\widetilde{c}(\mathbf{u}_i|\boldsymbol{\alpha})}{\sum_{j=1}^N \widetilde{c}(\mathbf{u_j}|\boldsymbol{\alpha})}\mathbf{u}_i\right]$ (Eq. 10) |
| ZARTS-GLD | $\mathbf{u}_i \sim \mathbb{S}^{d-1}$, a uniform distribution on a unit sphere. | $\mathbf{u}^* = \arg\min_i \{\mathcal{L}(\hat{\boldsymbol{\alpha}})|\hat{\boldsymbol{\alpha}} = \boldsymbol{\alpha}, \hat{\boldsymbol{\alpha}} = \boldsymbol{\alpha} + \mathbf{u}_i\}$ (Eq. 12) |

Fig. 1 (a). Consequently, we discard the first/second-order approximation in DARTS and instead use more accurate $\boldsymbol{\omega}^*$ coordinated with zero-order optimization.

Figure 1 (c) shows the optimization paths of DARTS and three methods of ZARTS, illustrating how the approximation in DARTS affects the search process. Starting from the same randomly generated position, we update architecture parameters $\boldsymbol{\alpha}$ for 10 iterations by DARTS and ZARTS and draw the tracks of the optimization path. ZARTS can gradually converge to the global minimum, while DARTS converges to an incorrect point.

## 4   ZERO-ORDER OPTIMIZATION FOR ARCHITECTURE SEARCH

This paper goes beyond the restrictive first/second-order approximation in DARTS and proposes to train architecture parameters $\boldsymbol{\alpha}$ in a zero-order optimization manner, allowing for more accurate estimation for $\boldsymbol{\omega}^*(\boldsymbol{\alpha})$. The generic form of our ZARTS framework is outlined in Alg. 1. Specifically, we select three representative methods, including a vanilla zero-order optimization algorithm, random search (RS) (Liu et al., 2020a), and two advanced algorithms: Maximum-likelihood Guided Parameter Search (MGS) (Welleck & Cho, 2020) and GradientLess Descent (GLD) (Golovin et al., 2020). They are presented in Sec. 4.1, 4.2, 4.3 also as preliminaries. Further, we theoretically establish the connection between ZARTS and DARTS, showing that ZARTS with RS and MGS optimizer can be seen as an expansion of DARTS. In the following, we use $\mathcal{L}(\boldsymbol{\alpha}) \triangleq \mathcal{L}_{val}(\boldsymbol{\omega}^*(\boldsymbol{\alpha}), \boldsymbol{\alpha})$ to denote the objective w.r.t. architecture parameters $\boldsymbol{\alpha} \in \mathbb{R}^d$ (Eq. 1), and $\mathcal{L}(\boldsymbol{\alpha} + \mathbf{u}) \triangleq \mathcal{L}_{val}(\boldsymbol{\omega}^*(\boldsymbol{\alpha} + \mathbf{u}), \boldsymbol{\alpha} + \mathbf{u})$, where $\mathbf{u}$ is the update for $\boldsymbol{\alpha}$.

### 4.1   ZARTS-RS VIA RANDOM SEARCH

Typical zero-order optimization methods have no access to first-order gradient information and construct gradient estimators based on zero-order information, typically, the function evaluation. As discussed in (Liu et al., 2020a), gradient estimation techniques can be categorized into different types based on the required number of function evaluations. *One-point gradient estimator* is one of the most popular algorithms (Liu et al., 2020a):

$$\hat{\nabla}_{\boldsymbol{\alpha}}\mathcal{L}(\boldsymbol{\alpha}) := \frac{\varphi(d)}{\mu}\mathcal{L}(\boldsymbol{\alpha} + \mu\mathbf{u})\mathbf{u}, \tag{4}$$

where $\mathbf{u} \sim q$ is sampled from a spherically symmetric distribution $q$, $\mu > 0$ is a smoothing parameter, and $\varphi(d)$ is a dimension-dependent factor related to $q$. Specifically, $q$ can either be a standard multivariate normal distribution $\mathcal{N}(\mathbf{0}, \mathbf{I})$ with $\varphi(d) = 1$, or a multivariate uniform distribution on a unit sphere $\mathbb{S}^{d-1}$ with $\varphi(d) = d$. Intuitively, ZARTS-RS samples nearby points from $\boldsymbol{\alpha}$ and yields a large updating step if the function value is high, consistent with the definition of the gradient.

*Two-point gradient estimator* is a natural extension of one-point estimator:

$$\hat{\nabla}_{\boldsymbol{\alpha}} \mathcal{L}(\boldsymbol{\alpha}) := \frac{\varphi(d)}{2\mu} \left[ \mathcal{L}(\boldsymbol{\alpha} + \mu\mathbf{u}) - \mathcal{L}(\boldsymbol{\alpha} - \mu\mathbf{u}) \right] \mathbf{u}. \tag{5}$$

We can also sample and average a batch of gradient estimations as Eq. 6 to reduce the variance of gradient estimators, resulting in the *multi-point estimator*:

$$\hat{\nabla}_{\boldsymbol{\alpha}} \mathcal{L}(\boldsymbol{\alpha}) := \frac{\varphi(d)}{2\mu N} \sum_{i=1}^{N} \left[ \mathcal{L}(\boldsymbol{\alpha} + \mu\mathbf{u}_i) - \mathcal{L}(\boldsymbol{\alpha} - \mu\mathbf{u}_i) \right] \mathbf{u}_i. \tag{6}$$

Following the first-order gradient descent algorithms, the architecture parameters $\boldsymbol{\alpha}$ are updated by $\boldsymbol{\alpha} \leftarrow \boldsymbol{\alpha} - \xi\hat{\nabla}_{\boldsymbol{\alpha}}\mathcal{L}(\boldsymbol{\alpha})$ with the learning rate $\xi$. Our implementation is compatible with all the estimators above, and we use multi-point estimator (Eq. 6) by default.

## 4.2 ZARTS-MGS VIA MAXIMUM-LIKELIHOOD GUIDED PARAMETER SEARCH

Maximum-likelihood guided parameter search (MGS) is an advanced zero-order optimization algorithm for machine translation (Welleck & Cho, 2020). We make the attempt to apply it to the NAS task. We first define a distribution for the update of architecture parameters, $\mathbf{u}$, as follows:

$$p(\mathbf{u}|\boldsymbol{\alpha}) = \frac{\widetilde{p}(\mathbf{u}|\boldsymbol{\alpha})}{Z(\boldsymbol{\alpha})} = \frac{1}{Z(\boldsymbol{\alpha})} \exp\left( -\frac{\mathcal{L}(\boldsymbol{\alpha} + \mathbf{u}) - \mathcal{L}(\boldsymbol{\alpha})}{\tau} \right), \tag{7}$$

where $\widetilde{p}(\mathbf{u}|\boldsymbol{\alpha}) = \exp\left(-[\mathcal{L}(\boldsymbol{\alpha} + \mathbf{u}) - \mathcal{L}(\boldsymbol{\alpha})]/\tau\right)$ is an unnormalized exponential distribution, and $Z(\boldsymbol{\alpha}) = \int \widetilde{p}(\mathbf{u}|\boldsymbol{\alpha})d\mathbf{u}$ is its normalization coefficient. $\tau$ is a temperature parameter controlling the variance of the distribution.

Intuitively, $\mathbf{u}$ with higher probability makes a more significant improvement on the objective function. Therefore, the optimal update of architecture parameters can be estimated by $\mathbf{u}^* = \mathbb{E}_{\mathbf{u} \sim p(\mathbf{u}|\boldsymbol{\alpha})}[\mathbf{u}]$. However, since the probability $p(\mathbf{u}|\boldsymbol{\alpha})$ is an implicit function relying on $\mathcal{L}(\boldsymbol{\alpha} + \mathbf{u})$, making it impractical to obtain the expectation, we refer to (Welleck & Cho, 2020) and apply importance sampling to sample from a proposal distribution $q(\mathbf{u}|\boldsymbol{\alpha})$ with known probability function:

$$\mathbf{u}^* = \mathbb{E}_{\mathbf{u} \sim p(\mathbf{u}|\boldsymbol{\alpha})}[\mathbf{u}] = \int \frac{\tilde{p}(\mathbf{u}|\boldsymbol{\alpha})}{Z(\boldsymbol{\alpha})} \mathbf{u} d\mathbf{u} = \int q(\mathbf{u}|\boldsymbol{\alpha}) \left[ \frac{\tilde{p}(\mathbf{u}|\boldsymbol{\alpha})}{Z(\boldsymbol{\alpha})q(\mathbf{u}|\boldsymbol{\alpha})} \mathbf{u} \right] d\mathbf{u}$$

$$= \mathbb{E}_{\mathbf{u} \sim q(\mathbf{u}|\boldsymbol{\alpha})} \left[ \frac{\tilde{p}(\mathbf{u}|\boldsymbol{\alpha})}{Z(\boldsymbol{\alpha})q(\mathbf{u}|\boldsymbol{\alpha})} \mathbf{u} \right] \approx \frac{1}{N} \sum_{i=1}^{N} \left[ \frac{\tilde{p}(\mathbf{u}_i|\boldsymbol{\alpha})}{Z(\boldsymbol{\alpha})q(\mathbf{u}_i|\boldsymbol{\alpha})} \mathbf{u}_i \right] \triangleq \hat{\mathbf{u}}^*, \tag{8}$$

where $\{\mathbf{u}_i\}_{i=1}^{N}$ are sampled from the proposal distribution $q(\mathbf{u}|\boldsymbol{\alpha})$. Similarly, the normalization coefficient $Z(\boldsymbol{\alpha})$ can be computed as follows:

$$Z(\boldsymbol{\alpha}) = \int \widetilde{p}(\mathbf{u}|\boldsymbol{\alpha})d\mathbf{u} \approx \frac{1}{N} \sum_{i=1}^{N} \left[ \frac{\tilde{p}(\mathbf{u}_i|\boldsymbol{\alpha})}{q(\mathbf{u}_i|\boldsymbol{\alpha})} \right]. \tag{9}$$

For convenience, we define a ratio representing the weight on each sample as $\widetilde{c}(\mathbf{u}|\boldsymbol{\alpha}) = \frac{\widetilde{p}(\mathbf{u}|\boldsymbol{\alpha})}{q(\mathbf{u}|\boldsymbol{\alpha})}$. Consequently, the optimal update for architecture parameters in Eq. 8 can be computed by:

$$\hat{\mathbf{u}}^* = \sum_{i=1}^{N} \left[ \frac{\widetilde{c}(\mathbf{u}_i|\boldsymbol{\alpha})}{\sum_{j=1}^{N} \widetilde{c}(\mathbf{u}_j|\boldsymbol{\alpha})} \mathbf{u}_i \right] = \sum_{i=1}^{N} \left[ \frac{\exp\left(-[\mathcal{L}(\boldsymbol{\alpha} + \mathbf{u}_i) - \mathcal{L}(\boldsymbol{\alpha})]/\tau\right)/q(\mathbf{u}_i|\boldsymbol{\alpha})}{\sum_{j=1}^{N} \exp\left(-[\mathcal{L}(\boldsymbol{\alpha} + \mathbf{u}_j) - \mathcal{L}(\boldsymbol{\alpha})]/\tau\right)/q(\mathbf{u}_j|\boldsymbol{\alpha})} \mathbf{u}_i \right]. \tag{10}$$

Finally, the architecture parameters are updated by $\boldsymbol{\alpha} \leftarrow \boldsymbol{\alpha} + \hat{\mathbf{u}}^*$. Additional importance sampling diagnostics are conducted to verify the effectiveness of ZARTS-MGS (see results in Appendix A.3).

### 4.3 ZARTS-GLD VIA GRADIENTLESS DESCENT

Unlike the above two algorithms that estimate gradient or the update for $\boldsymbol{\alpha}$, Golovin et al. (2020) propose the so-called GradientLess Descent (GLD) algorithm, which falls into the category of truly gradient-free (or direct search) methods. This work provides solid theoretical proof on the efficacy and efficiency of this approach and suggestions on the choice of search radius boundaries. Particularly, the authors prove the distance between the optimal minimum and the solution given by GLD is bounded and positively correlated with the condition number of the objective function, where the condition number $Q$ is defined as:

$$Q = \max_{1 \leq i \leq K} \left\{ \frac{|\mathcal{L}(\boldsymbol{\alpha} + \Delta_i) - \mathcal{L}(\boldsymbol{\alpha})| \cdot \|\boldsymbol{\alpha}\|}{\|\Delta_i\| \cdot |\mathcal{L}(\boldsymbol{\alpha})|} \right\}. \tag{11}$$

We notice the loss landscape w.r.t. $\boldsymbol{\alpha}$ is pretty flat, as shown in Fig. 1(b), implying a low condition number, thus the high efficiency of ZARTS-GLD.

Specifically, at each iteration, with a predefined search radius boundary $[r, R]$, we independently sample candidate updates $\{\mathbf{u}_i\}$ for architecture parameters on spheres with various radii $\{2^{-k}R\}_{k=0}^{\log(R/r)}$ and perform function evaluation at these points. By comparing $\mathcal{L}(\boldsymbol{\alpha})$ and $\{\mathcal{L}(\boldsymbol{\alpha} + \mathbf{u}_i)\}$, $\boldsymbol{\alpha}$ steps to the point with minimum value, or stay at the current point if none of them makes an improvement. The architecture parameters are then updated by $\boldsymbol{\alpha} \leftarrow \boldsymbol{\alpha} + \mathbf{u}^*$.

$$\mathbf{u}^* = \arg\min_i \{\mathcal{L}(\hat{\boldsymbol{\alpha}}) | \hat{\boldsymbol{\alpha}} = \boldsymbol{\alpha}, \hat{\boldsymbol{\alpha}} = \boldsymbol{\alpha} + \mathbf{u}_i\} \tag{12}$$

### 4.4 CONNECTION BETWEEN DARTS AND ZARTS

The similarity between gradient-estimation-based zero-order optimization and SGD builds an essential connection when the objective function is differentiable. Recall the smoothing parameter $\mu$ defined in Eq. 4, leading to the following definition of the smoothed version of $\mathcal{L}$:

$$\mathcal{L}_\mu(\boldsymbol{\alpha}) := \mathbb{E}_{\mathbf{u} \sim q'}[\mathcal{L}(\boldsymbol{\alpha} + \mu\mathbf{u})], \tag{13}$$

where $q' = \mathcal{N}(\mathbf{0}, \mathbf{I})$ if $q = \mathcal{N}(\mathbf{0}, \mathbf{I})$, and $q' = \mathbb{B}^d$ (a multivariate uniform distribution on a unit ball) if $q = \mathbb{S}^{d-1}$. The unbiasedness of Eq. 4 with respect to $\nabla_{\boldsymbol{\alpha}}\mathcal{L}_\mu(\boldsymbol{\alpha})$ is assured by:

$$\mathbb{E}_{\mathbf{u} \sim q}\left[\hat{\nabla}_{\boldsymbol{\alpha}}\mathcal{L}(\boldsymbol{\alpha})\right] = \nabla_{\boldsymbol{\alpha}}\mathcal{L}_\mu(\boldsymbol{\alpha}), \tag{14}$$

as proved by (Nesterov & Spokoiny, 2017; Berahas et al., 2021). The two-point and multi-point estimates have a similar unbiasedness condition when $\mathbb{E}_{\mathbf{u} \sim q}[\mathbf{u}] = 0$, which is satisfied in our case. The bias between $\hat{\nabla}_{\boldsymbol{\alpha}}\mathcal{L}(\boldsymbol{\alpha})$ and $\nabla_{\boldsymbol{\alpha}}\mathcal{L}(\boldsymbol{\alpha})$ is also bounded (Berahas et al., 2021; Liu et al., 2018b):

$$\mathbb{E}\left[\|\hat{\nabla}_{\boldsymbol{\alpha}}\mathcal{L}(\boldsymbol{\alpha}) - \nabla_{\boldsymbol{\alpha}}\mathcal{L}(\boldsymbol{\alpha})\|_2^2\right] = O(d)\|\nabla_{\boldsymbol{\alpha}}\mathcal{L}(\boldsymbol{\alpha})\|_2^2 + O\left(\frac{\mu^2 d^3 + \mu^2 d}{\varphi(d)}\right), \tag{15}$$

where $d, \mu, \varphi(d)$ have the same meanings as those in Sec. 4.1. Consequently, if $\mathcal{L}(\boldsymbol{\alpha})$ is indeed differentiable w.r.t $\boldsymbol{\alpha}$ and the iteration number $M$ is set to 1, ZARTS-RS degenerates to second-order DARTS with bounded error.

Next, we theoretically show that MGS (Welleck & Cho, 2020) will degenerate to gradient descent algorithm if the first-order Taylor approximation is applied. Then we analyze the relationship between ZARTS-MGS and DARTS.

**Proposition 1.** *Assuming that $\mathcal{L}(\boldsymbol{\alpha})$ in Eq. 1 is differentiable w.r.t. $\boldsymbol{\alpha}$, MGS algorithm (Welleck & Cho, 2020) degenerates to SGD (used in vanilla DARTS) by the first-order Taylor approximation for $\mathcal{L}(\boldsymbol{\alpha})$, i.e., $\mathbf{u}^* \propto -\nabla_{\boldsymbol{\alpha}}\mathcal{L}(\boldsymbol{\alpha})$.*

*Proof.* Denote $\mathbf{g} \triangleq \nabla_{\boldsymbol{\alpha}}\mathcal{L}(\boldsymbol{\alpha})$ as the gradient of $\mathcal{L}$. The Taylor series of $\mathcal{L}$ at $\boldsymbol{\alpha}$ up to the first order gives $\mathcal{L}(\boldsymbol{\alpha} + \mathbf{u}) - \mathcal{L}(\boldsymbol{\alpha}) \approx \mathbf{u}^\top \mathbf{g}$. Applying this approximation to the distribution of $\mathbf{u}$ (Eq. 7) yields:

$$p(\mathbf{u}|\boldsymbol{\alpha}) = \frac{e^{-\mathbf{u}^\top \mathbf{g}/\tau}}{Z(\mathbf{g})}, \; Z(\mathbf{g}) = \int_{\|\mathbf{u}\| \leq \varepsilon} e^{-\mathbf{u}^\top \mathbf{g}/\tau} d\mathbf{u}. \tag{16}$$

Here, the magnitude of $\mathbf{u}$ is constrained within $\varepsilon$ to make sure the rationality of first-order Taylor approximation. The optimal update $\mathbf{u}^*$ then becomes:

$$\mathbf{u}^* = \frac{\int_{\|\mathbf{u}\| \leq \varepsilon} \mathbf{u} \cdot e^{-\mathbf{u}^\top \mathbf{g}/\tau} d\mathbf{u}}{Z(\mathbf{g})} = -\frac{\nabla_{\mathbf{g}} Z(\mathbf{g})}{\tau Z(\boldsymbol{\alpha}, \mathbf{g})} = -\frac{1}{\tau} \nabla_{\mathbf{g}} \ln Z(\mathbf{g}). \tag{17}$$

Note that $\mathbf{u}^\top\mathbf{g} = -\|\mathbf{u}\|\|\mathbf{g}\|\cos\eta$, where $\eta$ is the angle between $\mathbf{u}$ and $\mathbf{g}$. According to the symmetry of integral, $Z(\mathbf{g})$ is determined once $\|g\|$ is given. Therefore, we can formulate $Z(\mathbf{g})$ as a function of $\|\mathbf{g}\|$: $Z(\mathbf{g}) = Z(\|\mathbf{g}\|)$. According to the chain rule:

$$\mathbf{u}^* = -\frac{1}{\tau}\nabla_\mathbf{g}\ln Z(\mathbf{g}) = -\frac{1}{\tau}\nabla_\mathbf{g}\ln Z(\|\mathbf{g}\|) = -\frac{\nabla_{\|\mathbf{g}\|}Z(\|\mathbf{g}\|)}{\tau Z(\|\mathbf{g}\|)}\nabla_g\|\mathbf{g}\| = -\frac{\nabla_{\|\mathbf{g}\|}Z(\|\mathbf{g}\|)}{\tau Z(\|\mathbf{g}\|)\|\mathbf{g}\|}\mathbf{g}. \quad (18)$$

Since $Z(\|\mathbf{g}\|), \nabla_{\|\mathbf{g}\|}Z(\|\mathbf{g}\|), \|\mathbf{g}\|$ are all scalars, we have

$$\mathbf{u}^* \propto -\mathbf{g} = -\nabla_\alpha\mathcal{L}(\alpha) = -\nabla_\alpha\mathcal{L}_{val}(\omega^*(\alpha), \alpha). \quad (19)$$

That is, the optimal update $\mathbf{u}^*$ in MGS algorithm shares a common direction with the negative gradient $-\nabla_\alpha\mathcal{L}(\alpha)$, as used by gradient descent. □

Based on Proposition 1, ZARTS-MGS can be seen as an expansion of DARTS, and it degrades to first-order and second-order DARTS when $\omega^*$ is estimated by $\omega^*_{1st}$ and $\omega^*_{2nd}$, respectively. In general, ZARTS-RS/-MGS can degenerate to DARTS, given the differentiability assumption.

However, unlike DARTS, which has to estimate $\omega^*(\alpha)$ by $\omega^*_{1st}$ or $\omega^*_{2nd}$ to satisfy the differentiablity property of $\mathcal{L}_{val}$ and update $\alpha$ by gradient descent algorithm, ZARTS, without such assumptions, can compute $\omega^*(\alpha)$ by training network weights $\omega$ for arbitrary numbers of iterations, leading to more robust and effective training for architecture parameters, as shown in the next section.

## 5 EXPERIMENTS

In this section, we first verify the stability of our ZARTS (with its three variants RS, MGS, GLD) on the four popular search spaces of R-DARTS (Zela et al., 2020b) on three datasets including CIFAR-10 (Krizhevsky et al., 2009), CIFAR-100 (Krizhevsky et al., 2009), and SVHN (Netzer et al., 2011). Note that though ZARTS-GLD performs best in Table 2, and it falls into the category of direct search methods, requiring more sampling for candidate updates $\mathbf{u}$ and thus more search cost (2.2 GPU-days on the search space of DARTS) than ZARTS-MGS (1.0 GPU-days). Considering the trade-off between accuracy and speed, ZARTS-MGS is chosen by default if not otherwise specified. We then follow Amended-DARTS (Bi et al., 2019) and empirically evaluate the convergence ability of our method by searching for 200 epochs. Performance trends of the discovered architectures along with the search process are drawn in Fig. 2(a). Finally, we search and evaluate on the search space of DARTS (Liu et al., 2019) to compare with peer methods and show the efficacy of our method. All the experiments are conducted on NVIDIA 2080Ti. The details of search spaces and experiment settings are given in Appendix B, due to space limitation.

### 5.1 STABILITY EVALUATION

The instability issue of gradient-based NAS methods has recently drawn increasing attention. Amended-DARTS (Bi et al., 2019) shows that after searching by DARTS for 200 epochs, skip-connection gradually dominates the discovered architectures. R-DARTS (Zela et al., 2020b) proposes four search spaces, S1-S4, which amplify the instability of DARTS, as such dominance occurs after only 50 epochs of searching. These studies expose the instability of gradient-based methods. To verify the stability of our method, we search on S1-S4 proposed by R-DARTS and conduct convergence analysis following Amended-DARTS.

Table 2: Test error (%) with DARTS and its variants on different search spaces. We adopt the same settings as R-DARTS (Zela et al., 2020a). The best and second best is underlined in boldface and in boldface, respectively.

| | | **DARTS** | **R-DARTS** | | **DARTS** | | **ZARTS (ours)** | | |
|---|---|---|---|---|---|---|---|---|---|
| | | | DP | L2 | ES | ADA | RS | MGS | GLD |
| CIFAR10 | S1 | 3.84 | 3.11 | 2.78 | 3.01 | 3.10 | 2.83 | **2.65** | **2.50** |
| | S2 | 4.85 | 3.48 | 3.31 | 3.26 | 3.35 | 3.35 | **3.24** | **3.08** |
| | S3 | 3.34 | 2.93 | **2.51** | 2.74 | 2.59 | 2.59 | **2.56** | **2.56** |
| | S4 | 7.20 | 3.58 | **3.56** | 3.71 | 4.84 | 4.90 | 3.70 | **3.52** |
| CIFAR100 | S1 | 29.46 | 25.93 | 24.25 | 28.37 | 24.03 | 23.64 | **23.16** | **23.33** |
| | S2 | 26.05 | 22.30 | 22.24 | 23.25 | 23.52 | 21.54 | **20.91** | **21.13** |
| | S3 | 28.90 | 22.36 | 23.99 | 23.73 | 23.37 | 22.62 | **22.33** | **21.90** |
| | S4 | 22.85 | 22.18 | 21.94 | **21.26** | 23.20 | 23.33 | 21.31 | **21.00** |
| SVHN | S1 | 4.58 | 2.55 | 4.79 | 2.72 | 2.53 | **2.40** | 2.51 | **2.48** |
| | S2 | 3.53 | 2.52 | 2.51 | 2.60 | 2.54 | 2.52 | **2.45** | **2.48** |
| | S3 | 3.41 | 2.49 | 2.48 | 2.50 | 2.50 | **2.41** | 2.52 | **2.44** |
| | S4 | 3.05 | 2.61 | 2.50 | 2.51 | **2.46** | 2.59 | **2.48** | 2.53 |

**Performance on S1-S4.** We first search on the four spaces on CIFAR-10, CIFAR-100, and SVHN. Our evaluation settings are the same as R-DARTS (Zela et al., 2020b). Specifically, for CIFAR-10,

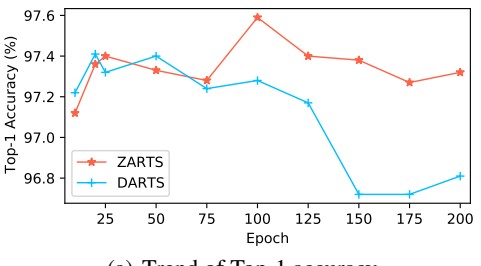
(a) Trend of Top-1 accuracy

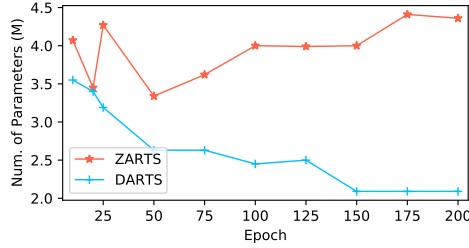
(b) Trend of Number of Parameters

Figure 2: Trends of accuracy and model size in the search process of DARTS and ZARTS for 200 epochs on CIFAR-10. The top-1 accuracy is obtained by training models for 600 epochs.

Table 3: Performance on CIFAR. The top block reports the accuracy of the best model. The bottom block gives the mean of four independent searches as recommended by (Zela et al., 2020b; Chen & Hsieh, 2020; Yu et al., 2020). ◇ Reported by (Dong & Yang, 2019). ⋆ by (Zela et al., 2020b).

| | CIFAR-10 | Params (M) ↓ | Error (%) ↓ | GPU Cost (days) ↓ |
|---|---|---|---|---|
| best | DARTS (1st) (2019) | 3.3 | 3.00 | 0.4 |
| | DARTS (2nd) (2019) | 3.4 | 2.76 | 1.0 |
| | P-DARTS (2019) | 3.4 | 2.50 | 0.3 |
| | GDAS (2019) | 3.4 | 2.93 | 0.2 |
| | ISTA-NAS (2020) | 3.3 | 2.54 | 0.05 |
| | PR-DARTS (2020) | 3.4 | 2.32 | 0.17 |
| | DARTS- (2021) | 3.5 | 2.50 | 0.4 |
| | **ZARTS (ours)** | 3.5 | **2.46** | 1.0 |
| average | DARTS(1st) (2020) | - | 3.38±0.23 | 0.4 |
| | MergeNAS (2020b) | 2.9 | 2.68±0.01 | 0.6 |
| | SGAS (Cri.2) (2020) | 3.9 | 2.67±0.21 | 0.3 |
| | R-DARTS (2020b) | - | 2.95±0.21 | 1.6 |
| | SDARTS-ADV (2020) | 3.3 | 2.61±0.02 | 1.3 |
| | Amended-DARTS (2019) | 3.3 | 2.71±0.09 | 1.7 |
| | DARTS- (2021) | 3.5±0.1 | 2.59±0.08 | 0.4 |
| | **ZARTS (ours)** | 3.7±0.3 | **2.54±0.07** | 1.0 |

| | CIFAR-100 | Params (M) ↓ | Error (%) ↓ | GPU Cost (days) ↓ |
|---|---|---|---|---|
| best | AmoebaNet (2019) | 3.1 | 18.93◇ | 3150 |
| | PNAS (2018a) | 3.2 | 19.53◇ | 150 |
| | ENAS (2018) | 4.6 | 19.43◇ | 0.45 |
| | P-DARTS (2019) | 3.6 | 17.49 | 0.3 |
| | GDAS (2019) | 3.4 | 18.38◇ | 0.2 |
| | ROME (2020a) | 4.4 | 17.33 | 0.3 |
| | PR-DARTS (2020) | 3.4 | 16.45 | 0.17 |
| | **ZARTS (ours)** | 4.0 | **15.46** | 1.0 |
| average | DARTS (2019) | - | 20.58±0.44⋆ | 0.4 |
| | R-DARTS (2020b) | - | 18.01±0.26⋆ | 1.6 |
| | ROME (2020a) | 4.4 | 17.41±0.12 | 0.3 |
| | DARTS- (2021) | 3.3 | 17.51±0.25 | 0.4 |
| | **ZARTS (ours)** | 4.1±0.13 | **16.29±0.53** | 1.0 |

the discovered models on S1 and S3 are constructed by 20 cells with 36 initial channels; models on S2 and S4 have 20 cells with 16 initial channels. For CIFAR-100 and SVHN, all the models on S1-S4 have 8 cells and 16 initial channels. Four parallel tests are conducted on each benchmark, among which the best is reported in Table 2. We observe that ZARTS achieves outstanding performance with great robustness on 12 benchmarks. All three zero-order algorithms outperform DARTS by a significant margin. Even the vanilla zero-order optimization algorithm ZARTS-RS achieves similar robust performance as R-DARTS, which verifies our analysis in Fig. 1, i.e., the coarse estimation $\omega_{1st}^*$ in DARTS distorts the landscape and causes instability. Additionally, to compare with SDARTS (Chen & Hsieh, 2020), we follow its experiment settings by increasing the number of cells and initial channels to 20 and 36 with results reported in Appendix B.4.

**Convergence Analysis.** The convergence ability of NAS methods describes whether a search method can stably discover effective architectures along the search process. For an effective NAS method with great convergence ability, the discovered architectures' ultimate performance (top-1 accuracy) should converge to a high value. Amended-DARTS (Bi et al., 2019) empirically shows that DARTS has a poor convergence ability: accuracy of the supernet increases but the ultimate performance of the searched network drops. Following Amended-DARTS, we run ZARTS and DARTS for 200 epochs and show the trend of performance and number of parameters in Fig. 2. Specifically, we derive one network every 25 epochs during the search process and train each network for 600 epochs to evaluate its ultimate performance. We observe that the networks searched by ZARTS perform stably well (around 97.40% accuracy), while the performance of networks searched by DARTS gradually drops. Moreover, the parameter number of networks searched by DARTS decreases significantly after 50 epochs, indicating that parameterless operations dominate the topology and the instability issue (Zela et al., 2020a) occurs. On the contrary, ZARTS consistently discovers effective networks with about 4.0M parameters, showing the great stability of our method.

## 5.2 COMPARISON WITH PEER METHODS ON THE SEARCH SPACE OF DARTS

To show the efficacy and effectiveness of our method, we search and evaluate on DARTS's search space on both CIFAR-10 and CIFAR-100. Additionally, the models searched on CIFAR-10 are transferred to ImageNet to evaluate the transferability of our method. The search space and settings follow DARTS (Liu et al., 2019), as is introduced in Appendix B.2.

**Results on CIFAR-10.** We conduct four parallel runs by searching with different random seeds and separately training the searched architectures for 600 epochs. The best and average accuracy of four parallel tests are reported in Table 3. In particular, ZARTS achieves 97.46% average accuracy and 97.54% best accuracy on CIFAR-10, outperforming DARTS and its variants. Also, compared with Amended-DARTS that approximates optimal operation weights $\omega^*(\alpha)$ by Hessian matrix, our method can stably discover effective architectures in fewer GPU days.

**Results on CIFAR-100.** Table 3 shows our method achieves 83.71% and 84.54% for mean and best accuracy (i.e. 1- error%), on CIFAR-100, outperforming the compared methods by more than 1%.

**Results on ImageNet.** For transferability test, we follow the settings of DARTS to transfer the network discovered on CIFAR-10 to ImageNet. Models are constructed by stacking 14 cells with 48 initial channels. We train 250 epochs with a batch size of 1024 by SGD with a momentum of 0.9 and a base learning rate of 0.5. We utilize the same data pre-processing strategies and auxiliary classifiers as DARTS. Table 4 shows the performance of our searched networks, wit two models evaluated. ZARTS (5.6M) has 5.6M parameters and achieves 75.7% top-1 accuracy on the validation set of ImageNet, and ZARTS (5.0M) has 5.0M parameters and achieves 75.5% accuracy. Their structure details are given in Appendix B.6, which has fewer skip connection operations than DARTS.

Table 4: Performance on ImageNet in DARTS's search space by two architectures. [†] direct search on ImageNet.

| Models | FLOPs (M) ↓ | Params (M) ↓ | Top-1 Err. (%) ↓ | GPU Cost (days) ↓ |
|---|---|---|---|---|
| AmoebaNet-A (2019) | 555 | 5.1 | 25.5 | 3150 |
| NASNet-A (2018) | 564 | 5.3 | 26.0 | 1800 |
| PNAS (2018a) | 588 | 5.1 | 25.8 | 225 |
| MdeNAS (2019) | - | 6.1 | 25.5 | 0.16 |
| DARTS (2nd) (2019) | 574 | 4.7 | 26.7 | 1.0 |
| P-DARTS (2019) | 557 | 4.9 | 24.4 | 0.3 |
| PC-DARTS (2020a) | 586 | 5.3 | 25.1 | 0.1 |
| FairDARTS-B (2020) | 541 | 4.8 | 24.9 | 0.4 |
| FairNAS-C[†] (2019) | 321 | 4.4 | 25.3 | 12 |
| SNAS (2019) | 522 | 4.3 | 27.3 | 1.5 |
| GDAS (2019) | 581 | 5.3 | 26.0 | 0.2 |
| SPOS[†] (2019) | 323 | 3.5 | 25.6 | 12 |
| ProxylessNAS[†] (2019) | 465 | 7.1 | 24.9 | 8.3 |
| FBNet-C[†] (2019) | 375 | 5.5 | 25.1 | 9 |
| Amended-DARTS (2019) | 586 | 5.2 | 24.7 | 1.7 |
| **ZARTS** (5.6M params) | 647 | 5.6 | **24.3** | 1.0 |
| **ZARTS** (5.0M params) | 573 | 5.0 | **24.5** | 1.0 |

## 6 FURTHER DISCUSSION AND CONCLUSION

The proposed ZARTS has opened new possible space for future work at least in the following aspects: i) We have incorporated the three adopted zero-order solvers in Table 1, which suggests new solvers may also be readily reused to improve ZARTS, e.g., in a way of AutoML that automatically determines the suited solver given specific tasks or datasets. In contrast, this feature is not allowed in DARTS as there is little option for the gradient-descent solver. ii) Since ZARTS can be seen as a gradient-free counterpart to DARTS, which in fact also requires the same exhaustive GPU memory as DARTS, memory-efficient techniques, e.g., single-path NAS ROME (Wang et al., 2020a) can also be adopted to reduce the memory cost as well as the computation cost in the search process.

DARTS has been a dominant paradigm in NAS, while its instability issue has received increasing attention (Bi et al., 2019; Zela et al., 2020b; Chen & Hsieh, 2020). In this work, we have empirically shown that the instability issue results from the first-order approximation for optimal network weights and the optimization gap in DARTS, which is also raised in the recent study (Bi et al., 2019). To step out of such a bottleneck, this work proposes a robust search framework named ZARTS, allowing for higher-order approximation for $\omega^*(\alpha)$ and supporting multiple combinations of zero-order optimization algorithms. Specifically, we adopt three representative methods for experiments and reveal the connection between ZARTS and DARTS. Extensive experiments on various benchmarks show the effectiveness and robustness of ZARTS. To our best knowledge, this is the first work that manages to apply zero-order optimization to one-shot NAS, which provides a promising paradigm to solve the bi-level optimization problem for NAS.

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

# A APPENDIX

## A.1 ESTIMATION FOR SECOND-ORDER PARTIAL DERIVATIVE IN DARTS

Liu et al. (2019) introduce second-order approximation to estimate optimal network weights, i.e., $\boldsymbol{\omega}^* \approx \boldsymbol{\omega}' = \boldsymbol{\omega} - \xi \nabla_{\boldsymbol{\omega}} \mathcal{L}_{train}(\boldsymbol{\omega}, \boldsymbol{\alpha})$, so that $\nabla_{\boldsymbol{\alpha}} \mathcal{L}_{val}(\boldsymbol{\omega}^*(\boldsymbol{\alpha}), \boldsymbol{\alpha}) \approx \nabla_{\boldsymbol{\alpha}} \mathcal{L}_{val}(\boldsymbol{\omega}', \boldsymbol{\alpha}) - \xi \nabla^2_{\boldsymbol{\alpha}, \boldsymbol{\omega}} \mathcal{L}_{train}(\boldsymbol{\omega}, \boldsymbol{\alpha}) \nabla_{\boldsymbol{\omega}'} \mathcal{L}_{val}(\boldsymbol{\omega}', \boldsymbol{\alpha})$. However, the second-order partial derivative is hard to compute, so the authors estimate it as follows:

$$\nabla^2_{\boldsymbol{\alpha}, \boldsymbol{\omega}} \mathcal{L}_{train}(\boldsymbol{\omega}, \boldsymbol{\alpha}) \nabla_{\boldsymbol{\omega}'} \mathcal{L}_{val}(\boldsymbol{\omega}', \boldsymbol{\alpha}) \approx \frac{\nabla_{\boldsymbol{\alpha}} \mathcal{L}_{train}(\boldsymbol{\omega}^+, \boldsymbol{\alpha}) - \nabla_{\boldsymbol{\alpha}} \mathcal{L}_{train}(\boldsymbol{\omega}^-, \boldsymbol{\alpha})}{2\epsilon}, \quad (20)$$

where $\boldsymbol{\omega}^{\pm} = \boldsymbol{\omega} \pm \epsilon \nabla_{\boldsymbol{\omega}'} \mathcal{L}_{val}(\boldsymbol{\omega}', \boldsymbol{\alpha})$, and $\epsilon = \frac{0.01}{\|\nabla_{\boldsymbol{\omega}'} \mathcal{L}_{val}(\boldsymbol{\omega}', \boldsymbol{\alpha})\|_2}$. Here, we prove that the above approximation in Eq. 20 is difference method.

*Proof.* First of all, to simplify the writing, we make the following definitions:

$$f(\boldsymbol{\omega}, \boldsymbol{\alpha}) = \nabla_{\boldsymbol{\alpha}} \mathcal{L}_{train}(\boldsymbol{\omega}, \boldsymbol{\alpha}), \quad g(\boldsymbol{\omega}, \boldsymbol{\alpha}) = \mathcal{L}_{val}(\boldsymbol{\omega}, \boldsymbol{\alpha}). \quad (21)$$

Then the left term in Eq. 20 can be simplified as:

$$\nabla^2_{\boldsymbol{\alpha}, \boldsymbol{\omega}} \mathcal{L}_{train}(\boldsymbol{\omega}, \boldsymbol{\alpha}) \nabla_{\boldsymbol{\omega}'} \mathcal{L}_{val}(\boldsymbol{\omega}', \boldsymbol{\alpha}) = \nabla_{\boldsymbol{\omega}} f(\boldsymbol{\omega}, \boldsymbol{\alpha}) \cdot \nabla_{\boldsymbol{\omega}'} g(\boldsymbol{\omega}', \boldsymbol{\alpha}) \quad (22)$$

$$= \nabla_{\boldsymbol{\omega}} f(\boldsymbol{\omega}, \boldsymbol{\alpha}) \cdot \frac{\nabla_{\boldsymbol{\omega}'} g(\boldsymbol{\omega}', \boldsymbol{\alpha})}{\|\nabla_{\boldsymbol{\omega}'} g(\boldsymbol{\omega}', \boldsymbol{\alpha})\|_2} \cdot \|\nabla_{\boldsymbol{\omega}'} g(\boldsymbol{\omega}', \boldsymbol{\alpha})\|_2 = \nabla_{\boldsymbol{\omega}} f(\boldsymbol{\omega}, \boldsymbol{\alpha}) \cdot \boldsymbol{l} \cdot \|\nabla_{\boldsymbol{\omega}'} g(\boldsymbol{\omega}', \boldsymbol{\alpha})\|_2, \quad (23)$$

where $\boldsymbol{l} = \frac{\nabla_{\boldsymbol{\omega}'} g(\boldsymbol{\omega}', \boldsymbol{\alpha})}{\|\nabla_{\boldsymbol{\omega}'} g(\boldsymbol{\omega}', \boldsymbol{\alpha})\|_2}$ is a unit vector. We notice $\nabla_{\boldsymbol{\omega}} f(\boldsymbol{\omega}, \boldsymbol{\alpha}) \cdot \boldsymbol{l}$ is the directional derivative of $f(\boldsymbol{\omega}, \boldsymbol{\alpha})$ along direction $\boldsymbol{l}$, which can be estimated by difference method with a small perturbation $\epsilon' = 0.01$:

$$\nabla_{\boldsymbol{\omega}} f(\boldsymbol{\omega}, \boldsymbol{\alpha}) \cdot \boldsymbol{l} \cdot \|\nabla_{\boldsymbol{\omega}'} g(\boldsymbol{\omega}', \boldsymbol{\alpha})\|_2 \approx \frac{f(\boldsymbol{\omega} + \epsilon' \boldsymbol{l}, \boldsymbol{\alpha}) - f(\boldsymbol{\omega} - \epsilon' \boldsymbol{l}, \boldsymbol{\alpha})}{2\epsilon'} \cdot \|\nabla_{\boldsymbol{\omega}'} g(\boldsymbol{\omega}', \boldsymbol{\alpha})\|_2 \quad (24)$$

Moreover, we define $\epsilon = \frac{\epsilon'}{\|\nabla_{\boldsymbol{\omega}'} g(\boldsymbol{\omega}', \boldsymbol{\alpha})\|_2}$. Then $\boldsymbol{\omega} \pm \epsilon' \boldsymbol{l} = \boldsymbol{\omega} \pm \epsilon \nabla_{\boldsymbol{\omega}'} g(\boldsymbol{\omega}', \boldsymbol{\alpha}) \triangleq \boldsymbol{\omega}^{\pm}$, so Eq. 24 can be simplified as:

$$\frac{f(\boldsymbol{\omega} + \epsilon' \boldsymbol{l}, \boldsymbol{\alpha}) - f(\boldsymbol{\omega} - \epsilon' \boldsymbol{l}, \boldsymbol{\alpha})}{2\epsilon'} \cdot \|\nabla_{\boldsymbol{\omega}'} g(\boldsymbol{\omega}', \boldsymbol{\alpha})\|_2 = \frac{f(\boldsymbol{\omega}^+, \boldsymbol{\alpha}) - f(\boldsymbol{\omega}^-, \boldsymbol{\alpha})}{2\epsilon}. \quad (25)$$

Substituting $f(\boldsymbol{\omega}, \boldsymbol{\alpha})$ in Eq. 21 with Eq. 25 results in Eq. 20. Therefore second-order approximation in DARTS utilizes difference method, which is also a zero-order optimization algorithm. $\square$

## A.2 LOSS LANDSCAPE W.R.T. ARCHITECTURE PARAMETERS

To draw loss landscapes w.r.t. $\boldsymbol{\alpha}$, we train a supernet for 50 epochs and randomly select two orthonormal vectors as the directions to perturb $\boldsymbol{\alpha}$. The same group of perturbation directions is used to draw landscapes for a fair comparison. Landscapes are plotted by evaluating $\mathcal{L}$ at grid points ranged from -1 to 1 at an interval of 0.02 in both directions. Fig. 3 illustrates landscapes (contours)

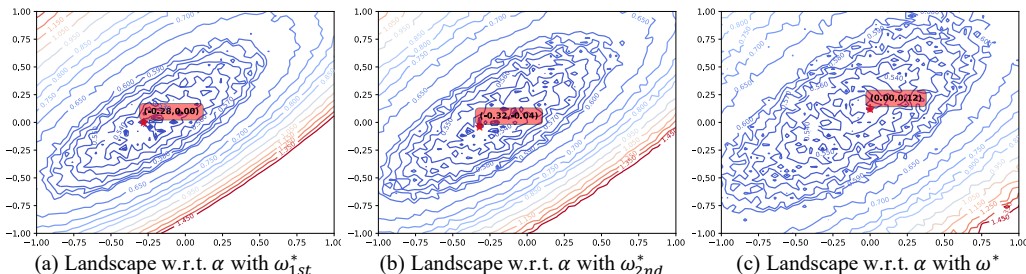

(a) Landscape w.r.t. $\alpha$ with $\omega^*_{1st}$     (b) Landscape w.r.t. $\alpha$ with $\omega^*_{2nd}$     (c) Landscape w.r.t. $\alpha$ with $\omega^*$

Figure 3: Loss landscapes w.r.t. architecture parameters $\boldsymbol{\alpha}$. In (a), we illustrate the landscape with first-order approximation. In (b), we illustrate the landscape with second-order approximation. In (c), we obtain $\boldsymbol{\omega}^*$ by training network weights $\boldsymbol{\omega}$ for 10 iterations, and illustrate the landscape w.r.t. $\boldsymbol{\alpha}$ with $\boldsymbol{\omega}^*$. To fairly compare the landscapes, we utilize the same model and candidate $\boldsymbol{\alpha}$ points. We observe the first/second-order approximations both sharpen the landscape.

w.r.t. $\boldsymbol{\alpha}$ under different order of approximation for optimal network weights, showing that both first- and second-order approximation sharpen the landscape and in turn lead to incorrect global minimum. In this work, we obtain $\boldsymbol{\omega}^*(\boldsymbol{\alpha})$ by fixing $\boldsymbol{\alpha}$ and fine-tuning network weights for $M$ iterations (Fig. 3 (c)). Selection of $M$ is also analyzed in Appendix B.3, showing that $M = 10$ iterations is accurate enough to estimate optimal network weights.

### A.3 DETAILS OF ZARTS-MGS ALGORITHM

**Selection of the Proposal Distribution $q(\mathbf{u}|\boldsymbol{\alpha})$.** Since the probability function of distribution $p$ is intractable, we sample from a proposal distribution $q$ and approximate the optimal update of architecture parameters by Eq. 10. The proposal distribution $q$ affects the efficiency of sampling. Specifically, an ideal $q$ should be as close to $p$ as possible when the sampling number is limited. Following Welleck & Cho (2020), we set the proposal distribution $q$ to a mixture of two Gaussian distributions, one of which is centered at the negative gradient of the loss function with current weights:

$$q(\mathbf{u}|\boldsymbol{\alpha}) = (1 - \lambda)\mathcal{N}(-\nabla_{\boldsymbol{\alpha}}\mathcal{L}_{val}(\boldsymbol{\omega}, \boldsymbol{\alpha}), \sigma^2) + \lambda\mathcal{N}(0, \sigma^2), \tag{26}$$

where $\sigma$ is the standard deviation. Intuitively, first-order DARTS (Liu et al., 2019) gives a hint: it updates the architecture parameters $\boldsymbol{\alpha}$ in the direction of $-\nabla_{\boldsymbol{\alpha}}\mathcal{L}_{val}(\boldsymbol{\omega}, \boldsymbol{\alpha})$. The gradient is an imperfect but workable direction with easy access.

**Importance Sampling Diagnostics.** To demonstrate that importance sampling and our choice of the proposal distribution is indeed appropriate in our case, we evaluate the effectiveness of $q$ (the proposal distribution) quantitatively by the following experiments. According to Owen (2013), effective sample size (ESS) $N_e$ is a popular indicator defined as $N_e = \frac{1}{\sum_{i=1}^{N} c_i^2}$. For $N$ samples, a larger $N_e \in [1, N]$ usually indicates a more effective sampling. On the contrary, small $N_e$ implies imbalanced sample weights and therefore is unreliable (Owen, 2013).

To evaluate the effectiveness of sampling in ZARTS-MGS, we set the sampling number $N$ to various values and plot $N_e$ versus epochs in each case. As is shown in Fig. 4(a), $N_e$ gradually approaches $N$ in all settings, indicating that different sampling numbers in our setting are meaningful, including larger ones (otherwise $N_e$ may "saturate").

As a further exploration, we define effective sample ratio (ESR) $R_e$ as the ratio of effective sampling to all samples:

$$R_e = \frac{N_e}{N}. \tag{27}$$

The value of $R_e$ denotes the bias between the target distribution $p$ and proposal distribution $q$, and a smaller value indicates a greater difference. $R_e$ is plotted against epochs for various $N$ in Fig. 4(b). On the one hand, $R_e$ at epoch 50 stabilizes at 0.7 as $N$ increases, which is an acceptable level of bias between $p$ and $q$ and supports our choice of $q$; on the other hand, we notice $R_e$ has already converged when $N \geq 4$. Considering the trade-off between estimation accuracy and speed, we set $N = 4$ as default, which is further discussed in Appendix B.3.

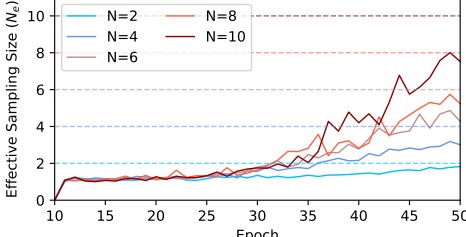
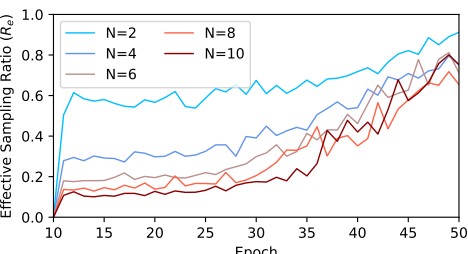

(a) ESS $N_e$ versus epochs over sampling number $N$.   (b) ESR $R_e$ versus epochs over sampling number $N$.

Figure 4: ESS $N_e$ and ESR $R_e$ versus epochs with different sampling numbers $N$ in the search stage on CIFAR-10. We fix iteration number $M = 10$ for all settings.

# B    SUPPLEMENTARY EXPERIMENTS

## B.1    DETAILS OF SEARCH SPACES

**DARTS's Standard Search Space.** The operation set $\mathcal{O}$ contains 7 basic operations: skip connection, max pooling, average pooling, $3 \times 3$ separable convolution, $5 \times 5$ separable convolution, $3 \times 3$ dilated separable convolution, and $5 \times 5$ dilated separable convolution. Though zero operation is included in the origin search space of DARTS (Liu et al., 2019), it will never be selected in the searched architecture. Therefore, we remove zero operation from the search space. We search and evaluate on the CIFAR-10 (Krizhevsky et al., 2009) dataset in this search space, and then transfer the searched model to ImageNet (Deng et al., 2009). Additionally, in our convergence analysis, we search by DARTS and ZARTS in this search space for 200 epochs.

**RDARTS's Search Spaces S1-S4.** To evaluate the stability of search algorithm, RDARTS (Zela et al., 2020a) designs four search spaces where DARTS suffers from instability severely, i.e. normal cells are dominated by parameter-less operations (such as identity and max pooling) after searching for 50 epochs. In S1, each edge in the supernet only has two candidate operations, but the candidate operation set for each edge differs; in S2, the operation set $\mathcal{O}$ only contains $3 \times 3$ separable convolution and identity for all edges; in S3, $\mathcal{O}$ contains $3 \times 3$ separable convolution, identity, and zero for all edges; in S4, $\mathcal{O}$ contains $3 \times 3$ separable convolution and noise operation for all edges. Please refer to Zela et al. (2020a) for more details of the search spaces.

## B.2    EXPERIMENT SETTINGS

**Search Settings.** Similar to DARTS, we construct a supernet by stacking 8 cells with 16 initial channels. We apply Alg. 1 to train architecture parameters $\boldsymbol{\alpha}$ for 50 epochs. Two hyper-parameters of ZARTS, sampling number $N$ and iteration number $M$, are set to 4 and 10 respectively. Ablation studies of the two hyper-parameters are analyzed in Appendix B.3. The setup for training $\boldsymbol{\omega}$ follows DARTS: SGD optimizer with a momentum of 0.9 and a base learning rate of 0.025. Our experiments are conducted on NVIDIA 2080Ti. ZARTS-MGS algorithm is used in supplementary experiments by default.

**Evaluation Settings.** We follow DARTS (Liu et al., 2019) and construct models by stacking 20 cells with 36 initial channels. Models are trained for 600 epochs by SGD with a batch size of 96. Cutout and auxiliary classifiers are used as introduced by DARTS.

## B.3    ABLATION STUDIES

There are two hyper-parameters in our method: sampling number $N$ and iteration number $M$. As introduced in Section 4, $N$ samples of update step of architecture parameters $\mathbf{u}_i$ are drawn to estimate the optimal update $\mathbf{u}^*$. For each sampled $\mathbf{u}_i$, we approximate the optimal weights $\boldsymbol{\omega}^*(\boldsymbol{\alpha} + \mathbf{u}_i)$ for each sample by training $\boldsymbol{\omega}$ for $M$ iterations. To evaluate the sensitivity of our method to the two hyper-parameters above, we conduct ablation studies on the standard search space of DARTS on CIFAR-10 dataset.

**Sensitivity to the Sampling Number $N$.** For various sampling numbers $N$, the average performance of three parallel searches with different random seeds is reported in Table 5. In this experiment, iteration number $M$ is fixed to 10. When $N = 2$, ZARTS achieves 97.37% accuracy with 2.87M parameters. The number of parameters of searched network increases as $N$ increases, and the performance of searched network gets stable when $N \geq 4$. Our method performs better than DARTS when $N = 4$, with similar search cost (1.0 GPU days). When $N = 6$, ZARTS achieves its best accuracy (97.49%) and costs 1.1 GPU days. When $N$ continues to increase, our method attempts to find more complex architectures (with 4.31M and 4.26M parameters).

Table 5: Comparison of different sampling numbers to approximate the optimal update for architecture parameters on the standard search space of DARTS on CIFAR-10 dataset. For each $N$, three parallel tests are conducted by searching on different random seeds and the mean and standard deviation of top-1 accuracy are reported.

| Sampling number | $N$=2 | $N$=4 | $N$=6 | $N$=8 |
|---|---|---|---|---|
| Error (%) | 2.63 | 2.54 | 2.51 | 2.57 |
| STD | ±0.12 | ±0.07 | ±0.09 | ±0.11 |
| Params (M) | 2.87 | 3.71 | 3.53 | 4.31 |
| Cost (GPU days) | 0.5 | 1.0 | 1.1 | 1.5 |

Table 7: Comparison with peer methods under the settings of SDARTS. (*left*) Test error of other methods are obtained from SDARTS (Chen & Hsieh, 2020), indicating the best performance among four replicate experiments with different random seeds. Note that 'RS' in SDARTS indicates random smoothing technique, while 'RS' in ZARTS indicates random search, a zero-order optimization algorithm. The best and second best is underlined in boldface and in boldface, respectively. (*right*) We report the average error and standard deviation of our method among four replicate experiments. Since PC-DARTS and SDARTS do not provide the average performance, we only compare the three implementations of our ZARTS.

| | | PC-DARTS | SDARTS | | ZARTS | | | ZARTS (avg.±std) | | |
| --- | --- | --- | --- | --- | --- | --- | --- | --- | --- | --- |
| | | | RS | ADV | RS | MGS | GLD | RS | MGS | GLD |
| CIFAR10 | S1 | 3.11 | 2.78 | 2.73 | 2.83 | **2.65** | **2.50** | 3.10±0.20 | 2.79±0.14 | **2.73±0.07** |
| | S2 | 3.02 | 2.75 | 2.65 | **2.41** | **2.39** | 2.60 | **2.60±0.14** | 2.65±0.17 | 2.67±0.08 |
| | S3 | **2.51** | 2.53 | **2.49** | 2.59 | 2.56 | 2.56 | 2.89±0.30 | 2.74±0.10 | **2.70±0.09** |
| | S4 | 3.02 | 2.93 | 2.87 | 3.35 | **2.74** | **2.63** | 4.11±1.07 | **2.99±0.21** | 3.35±0.93 |
| CIFAR100 | S1 | 18.87 | 17.02 | **16.88** | **17.38** | 17.62 | 17.40 | 18.07±0.55 | 18.20±0.48 | **17.83±0.44** |
| | S2 | 18.23 | 17.56 | 17.24 | **16.05** | **16.41** | 16.69 | **17.04±0.70** | 17.35±0.75 | 17.14±0.39 |
| | S3 | 18.05 | 17.73 | 17.12 | 17.22 | **17.03** | **16.58** | 18.14±0.72 | 17.72±0.46 | **16.99±0.38** |
| | S4 | 17.16 | 17.17 | **15.46** | 18.23 | 16.57 | **15.97** | 19.08±1.01 | 17.33±0.73 | **16.63±0.75** |
| SVHN | S1 | 2.28 | 2.26 | 2.16 | **2.09** | **2.13** | 2.14 | 2.21±0.10 | **2.17±0.03** | 2.20±0.04 |
| | S2 | 2.39 | 2.37 | 2.07 | **2.06** | **2.06** | 2.15 | 2.16±0.10 | **2.10±0.03** | 2.22±0.07 |
| | S3 | 2.27 | 2.21 | **2.05** | 2.17 | 2.20 | **2.07** | 2.27±0.14 | 2.25±0.05 | **2.13±0.07** |
| | S4 | 2.37 | 2.35 | **1.98** | 2.49 | **2.04** | 2.15 | 2.56±0.09 | **2.20±0.11** | 2.26±0.09 |

**Sensitivity to the Iteration Number $M$.** DARTS and its variants (Xu et al., 2020b; Zela et al., 2020b) assume that the optimal operation weights $\omega^*(\alpha)$ is differentiable w.r.t. architecture parameters $\alpha$, which has not been theoretically proved. In this work, we relax the above assumption and adopt zero-order optimization to update $\alpha$. As introduced in Section 4, we perform multiple iterations of gradient descent on operation

Table 6: Comparison of different iteration numbers to approximate the optimal operation weights in DARTS's standard search space on CIFAR-10. We conduct three parallel tests for each $M$ and report the mean and standard deviation of top-1 accuracy.

| Iteration number | $M$=2 | $M$=5 | $M$=8 | $M$=10 |
| --- | --- | --- | --- | --- |
| Error (%) | 2.62 | 2.60 | 2.57 | 2.54 |
| STD | ±0.15 | ±0.09 | ±0.03 | ±0.07 |
| Params (M) | 2.91 | 3.40 | 3.52 | 3.71 |

weights to accurately estimate $\omega^*(\alpha)$. To further confirm our analysis on the impact of iteration number $M$, we search with various values of $M$ and report the average performance of three parallel searches with different random seeds in Table 6. In this experiment, we set the sampling number $N$ to 4. The results reveal that the performance of searched model improves as $M$ increases and the highest accuracy is achieved at $M = 10$, which supports our analysis that inaccurate estimation for optimal operation weights $\omega^*(\alpha)$ can mislead the search procedure.

## B.4 COMPARISION WITH PEER METHODS ON S1-S4

Unlike R-DARTS (Zela et al., 2020b) that constructs models by stacking 8 cells and 16 inital channels, SDARTS (Chen & Hsieh, 2020) builds models by stacking 20 cells and 36 initial channels. To compare with SDARTS for fair, we follow its settings and report our results in Table 7. Specifically, we conduct four parallel tests on each benchmark by searching with different random seeds. Tabel 7 reports the best and average performance of our method. Note that other methods in Tabel 7 only report the best performance of four parallel tests. According to the results, we observe our ZARTS achieves state-of-the-art on 7 benchmarks and SDARTS-ADV slightly outperforms our ZARTS on 5 benchmarks.

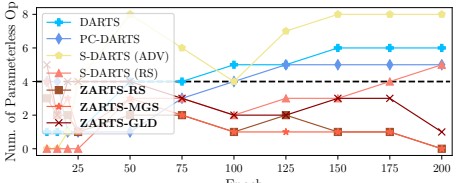 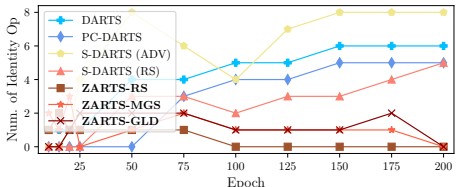

(a) Trend of Number of Parameterless Operations  (b) Trend of Number of Identity Operations

Figure 5: Trends of number of parameterless operations and identity operations in each normal cell searched by different NAS methods on CIFAR-10 for 200 epochs. The parameterless operations include max pooling, average pooling, and identity operation.

## B.5 CONVERGENCE ANALYSIS

The convergence ability of NAS methods describes whether a search method can stably discover effective architectures along the search process. A robust and effective NAS method should be able to converge to exemplary architectures with high performance. This work follows Amended-DARTS (Bi et al., 2019) and evaluates the convergence ability by searching for an extended period (200 epochs). However, since it is time-consuming to train every derived architecture along the search process, we illustrate the trend of number of parameterless operations (pooling and identity operations) in each normal cell to represent the performance of architectures (Fig. 5). Recent works (Chen et al., 2019; Bi et al., 2019; Zela et al., 2020b) show that architecture with more than 4 parameterless operations (especially identity operations) usually has a bad performance, which is a typical phenomenon of the instability issue. Here, we show the number of parameterless operations of our ZARTS in Fig. 5 and compare with another three methods, including DARTS, PC-DARTS and S-DARTS. We observe that the architectures discovered by DARTS, PC-DARTS and S-DARTS will be gradually dominated by parameterless operations (especially identity operation), implying that the instability issue occurs. In contrast, our ZARTS can stably control the number of parameterless operations.

## B.6 VISUALIZATION OF ARCHITECTURES

Note that in all our experiments, we directly utilize the architecture at the final epoch (epoch 50) as the inferred network. No model selection procedure is needed.

We visualize the architectures of normal and reduction cells searched by ZARTS in DARTS's search space on CIFAR-10, as is shown in Fig. 6. The architecture searched on CIFAR-100 dataset is illustrated in Fig. 7. We also conduct experiments in the four difficult search spaces of RDARTS (Zela et al., 2020a) on CIFAR-10 (Krizhevsky et al., 2009), CIFAR-100 (Krizhevsky et al., 2009), and SVHN (Netzer et al., 2011). The searched architectures are illustrated in Fig. 8, Fig. 9, and Fig. 10.

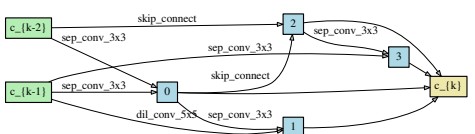 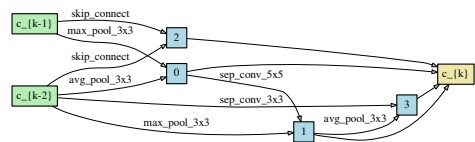

(a) The normal cell searched by ZARTS  (b) The reduction cell searched by ZARTS

Figure 6: The architectures of normal and reduction cell searched by ZARTS on CIFAR-10 in DARTS's search space. Model constructed by the above cells achieves $97.54\%$ accuracy on the CIFAR-10 dataset with 3.5M parameters.

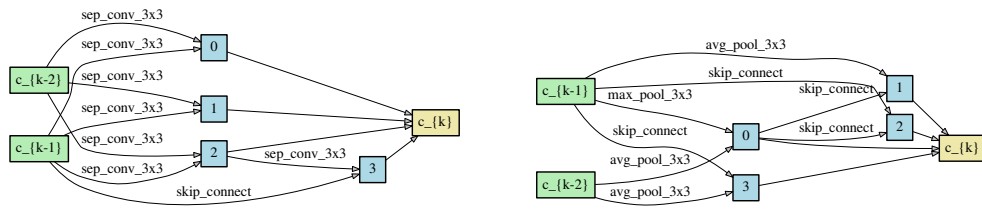

(a) The normal cell searched by ZARTS     (b) The reduction cell searched by ZARTS

Figure 7: The architectures of normal and reduction cell searched by ZARTS on CIFAR-100 in DARTS's search space. Model constructed by the above cells achieves $84.54\%$ accuracy on the CIFAR-100 dataset with $4.0$M parameters.

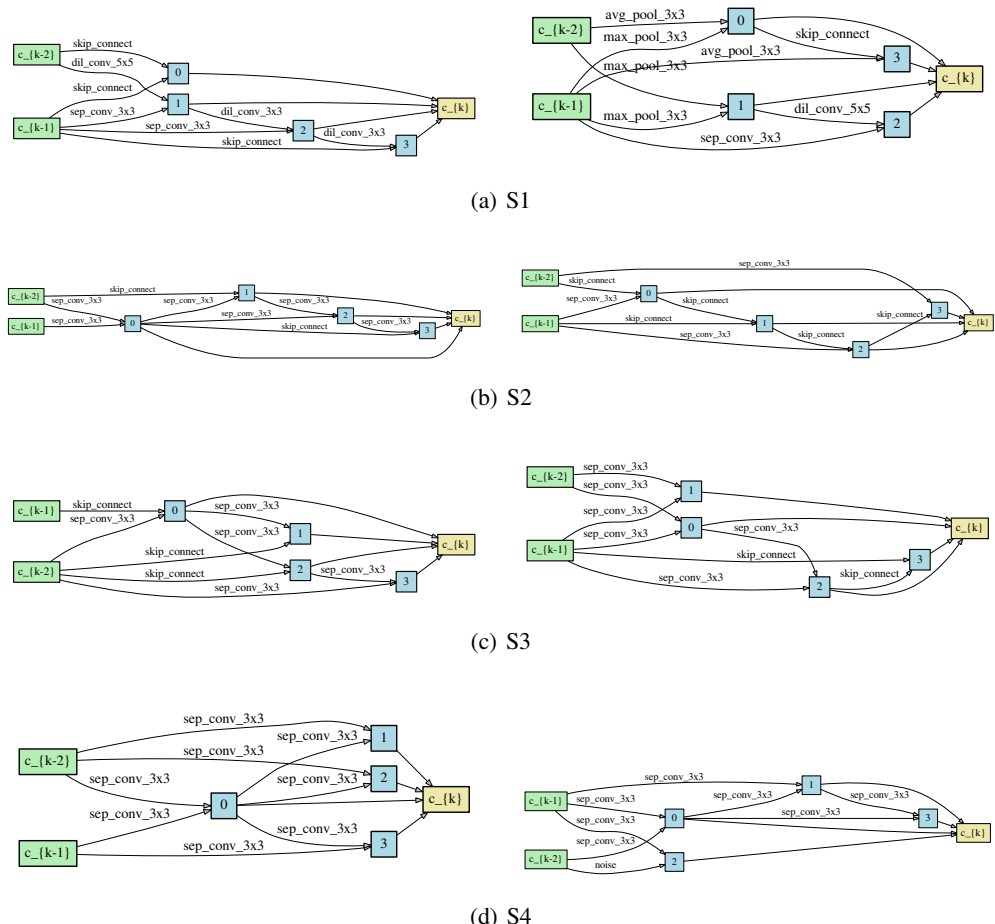

(a) S1

(b) S2

(c) S3

(d) S4

Figure 8: The architectures of normal and reduction cells searched by ZARTS on CIFAR-10 in the four difficult search space of RDARTS. The left column shows the normal cells, while the right column shows the reduction cells.

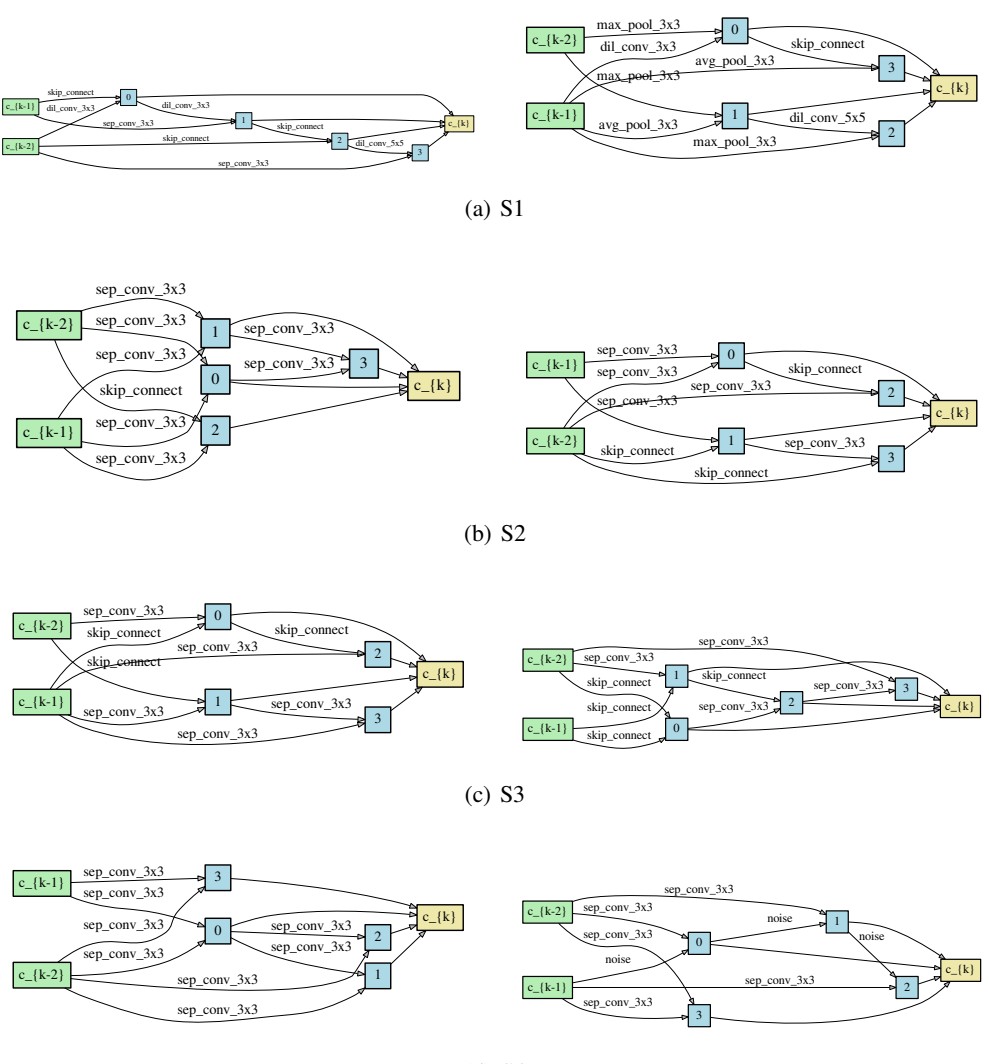

Figure 9: The architectures of normal and reduction cells searched by ZARTS on CIFAR-100 in the four difficult search space of RDARTS. The left column shows the normal cells, while the right column shows the reduction cells.

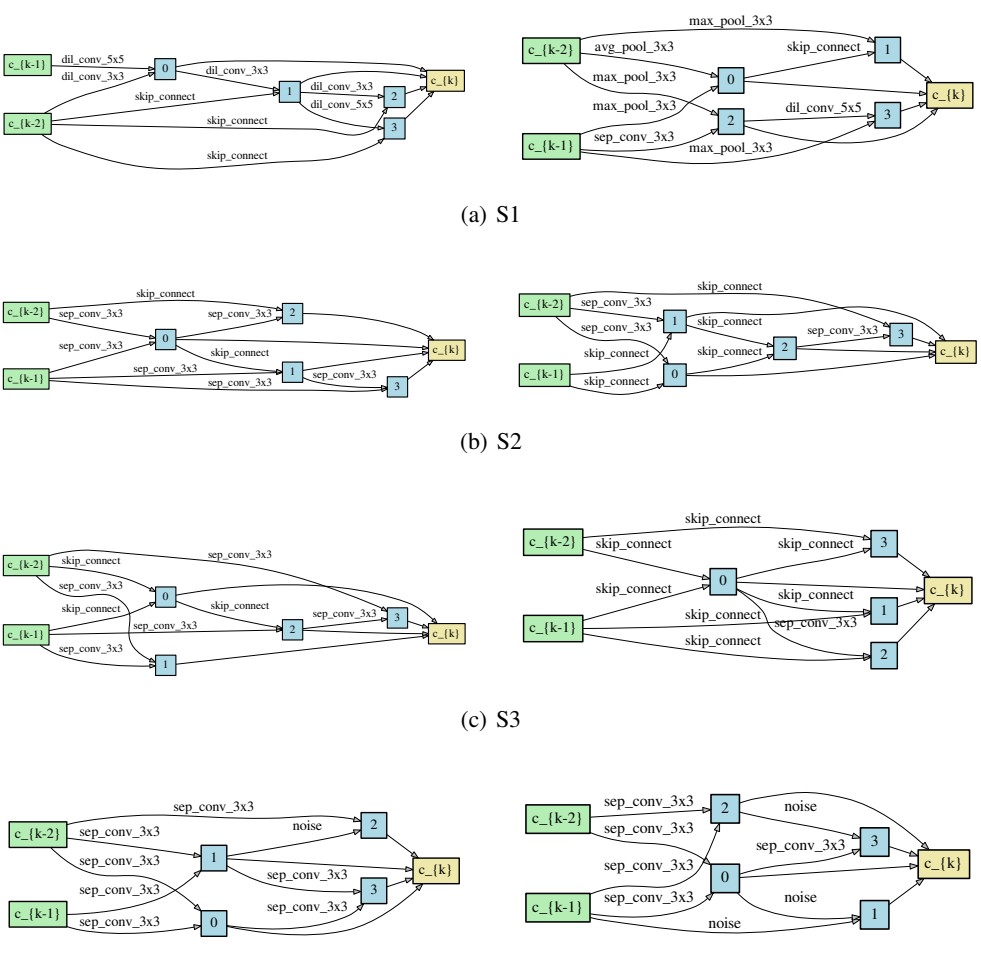

(a) S1

(b) S2

(c) S3

(d) S4

Figure 10: The architectures of normal and reduction cells searched by ZARTS on SVHN in the four difficult search space of RDARTS. The left column shows the normal cells, while the right column shows the reduction cells.

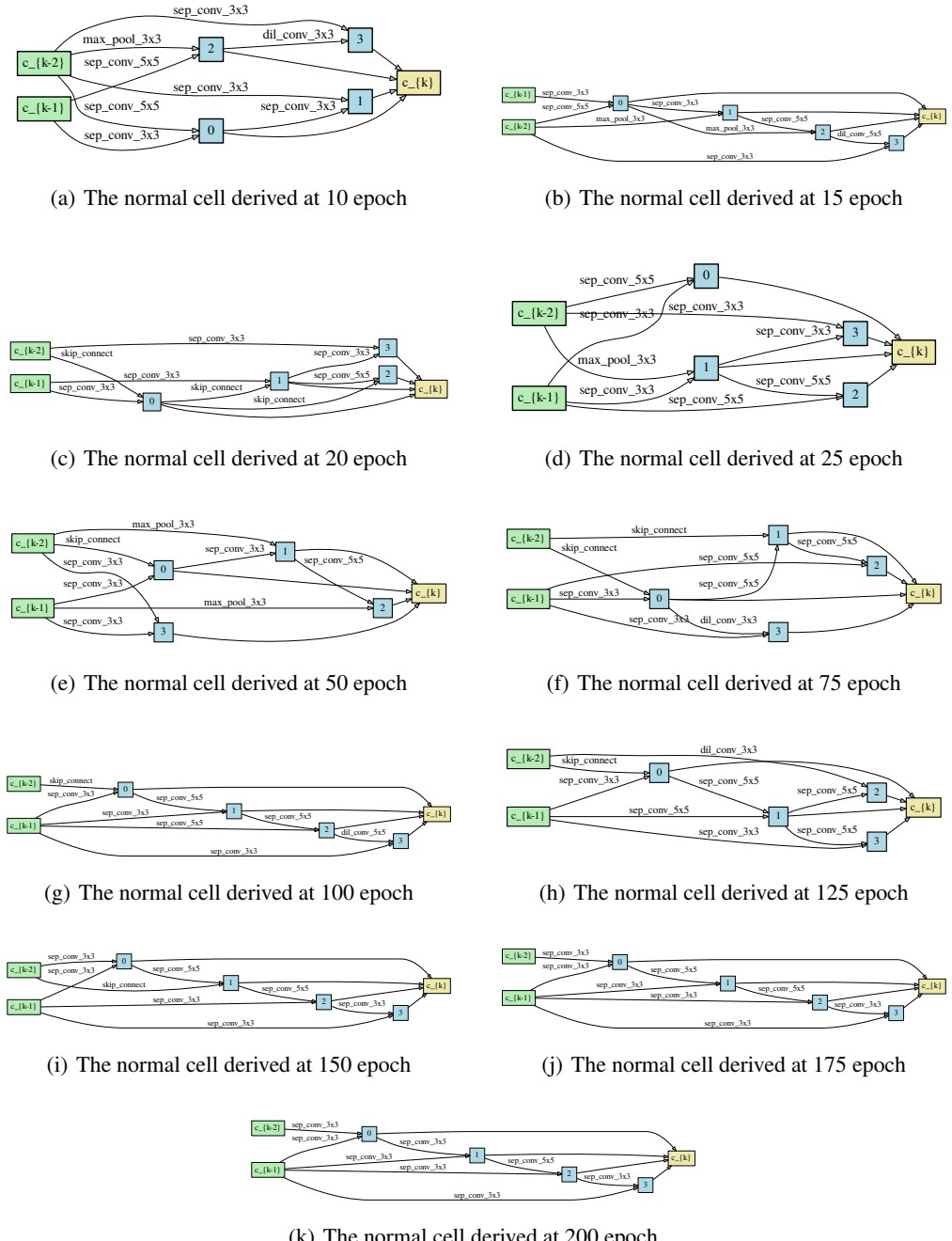

(a) The normal cell derived at 10 epoch

(b) The normal cell derived at 15 epoch

(c) The normal cell derived at 20 epoch

(d) The normal cell derived at 25 epoch

(e) The normal cell derived at 50 epoch

(f) The normal cell derived at 75 epoch

(g) The normal cell derived at 100 epoch

(h) The normal cell derived at 125 epoch

(i) The normal cell derived at 150 epoch

(j) The normal cell derived at 175 epoch

(k) The normal cell derived at 200 epoch

Figure 11: The derived architectures of normal cell every 25 epochs, which are searched by ZARTS on CIFAR-10 for 200 epochs.

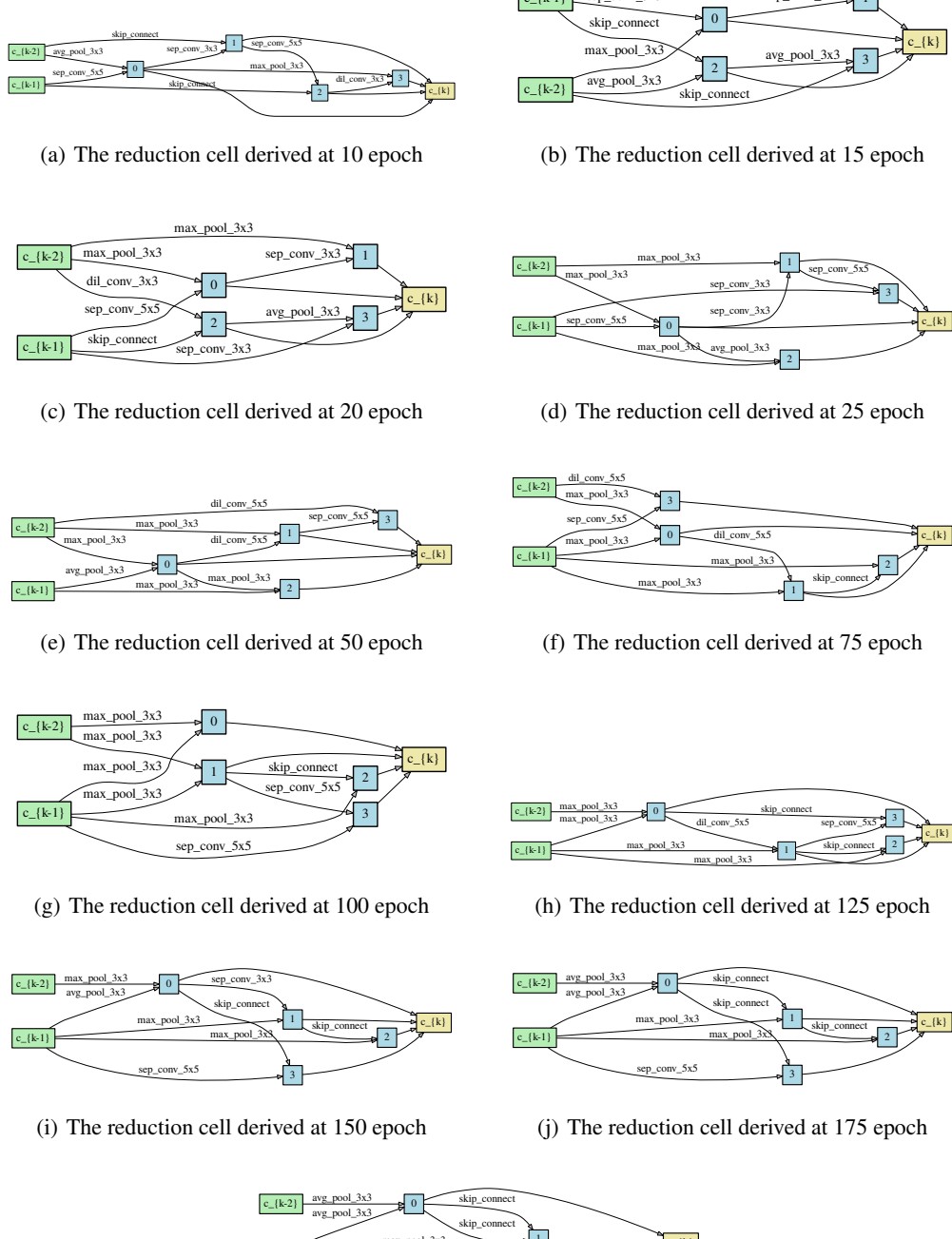

(a) The reduction cell derived at 10 epoch

(b) The reduction cell derived at 15 epoch

(c) The reduction cell derived at 20 epoch

(d) The reduction cell derived at 25 epoch

(e) The reduction cell derived at 50 epoch

(f) The reduction cell derived at 75 epoch

(g) The reduction cell derived at 100 epoch

(h) The reduction cell derived at 125 epoch

(i) The reduction cell derived at 150 epoch

(j) The reduction cell derived at 175 epoch

(k) The reduction cell derived at 200 epoch

Figure 12: The derived architectures of reduction cell every 25 epochs, which are searched by ZARTS on CIFAR-10 for 200 epochs.

