# OpenReview forum: "ZARTS: On Zero-order Optimization for Neural Architecture Search"
_ICLR.cc/2022/Conference — ICLR 2022 Submitted_

### Official Review · Reviewer_5Kkb · 2021-10-19

**Correctness:** 4
**Technical Novelty And Significance:** 2
**Empirical Novelty And Significance:** 2
**Recommendation:** 6
**Confidence:** 4

**Main Review:**

Pros
++ The analysis of how first-order optimization sharpens the landscape of loss (fig 1) is interesting. It sheds some lights on why first-order DARTS faces the problem of instability during the search.
++ The paper is the first to apply zero-order optimization methods to solve NAS, which could open a new research direction of NAS.
++ The analysis on connecting ZARTS and DARTS provides a theoretical explanation on why ZARTS can work better than DARTS.
++ The experiments are extensive and can demonstrate the effectiveness of the proposed ZARTS.

Cons
-- The three zero-order optimization methods are all proposed by other works, which weakens the novelty of this paper.
-- The search costs of ZARTS are not appealing comparing to other differentiable NAS methods like P-DARTS or FairDARTS (Table 4). Moreover, the top-1 error rate of ZARTS is similar to that of P-DARTS, while being slower at search.
- The superiority of ZARTS is not obvious in Table 7 comparing to SDARTS.

**Summary Of The Paper:**

This paper (ZARTS) proposes to apply gradient-estimation-based zero-order optimization methods to tackle neural architecture search (NAS). Two major contributions are made by this paper: (1) it is the first to borrow the methods of zero-order optimization to solve NAS  problem; (2) it shows that zero-order optimization methods can greatly avoid the training instability of first- or second-order optimization NAS methods like DARTS (which sharpens the loss landscape). Moreover, this paper also provides an explanation on how ZARTS is connected with DARTS. Extensive experiments demonstrate the efficacy of ZARTS.

**Summary Of The Review:**

Overall, it is a good paper that opens a new research direction of differentiable NAS family. The quality in terms of visualization, theoretical proof and empirical evaluation are also good. So, I recommend an 'accept' for this paper. The concern preventing me from giving higher rating is that all the optimization methods are directly obtained from previous works.

---

> ### Author Response · Authors · 2021-11-19
> **Thank you for your feedback**
>
> Thank you for the time and thorough feedback. Please see our general response and the detailed reply below, which we hope will resolve your concern about the novelty and effectiveness of our method.
>
>
> >**Q1:** Novelty of ZARTS.
>
> **A1:** Please see our general response. We hope our response can help to address your concern about the novelty of our method.
>
> In this work, we attempt to point out the harm of inaccurate approximation for optimal network weights ${\omega}^*({\alpha})$, which has been neglected in the literature.
> Specifically, We reveal that first-order approximation can distort the search loss landscape (analyzed in Sec. 3.2) and introduce an efficient search method ZARTS based on zero-order optimization algorithms that can robustly discover architectures in 1.0 GPU-day. Extensive analysis and experiments verify the effectiveness of our method.
>
> Moreover, we hope this paper can provide a new perspective of zero-order optimization to the NAS area and shed light on the cause of the instability issue in DARTS.
> However, designing a brand-new zero-order optimization algorithm for general cases is out of the scope of this paper, which we are willing to explore in the future.
>
> >**Q2:** The search cost is high.
>
> **A2:** Please refer to our general response (Computational Cost), which we hope will resolve your concern.
>
> >**Q3:** The improvement of accuracy compared to SDARTS is not significant.
>
> **A3:** Please refer to our general response (Performance Advantage), which we hope will resolve your concern.
>
> As for the comparison with S-DARTS, we compare three variants of ZARTS with it on the four search spaces in Table 7, showing that ZARTS achieves state-of-the-art on 7 benchmarks and SDARTS-ADV slightly outperforms ours on another 5 benchmarks.
> Moreover, we add new experimental results and analysis on convergence ability to the Appendix (Fig. 5 and Sec. B.5), which compares our method with S-DARTS, DARTS, and PC-DARTS, showing that our ZARTS has the best convergence ability.

---

> > ### Comment · Reviewer_5Kkb · 2021-11-26
> > **Post-rebuttal**
> >
> > Thanks for the responses to my concerns. Although the search costs are relatively high comparing to other works that focus on the speed-up of the search, I think that this work explores a new way, i.e. zero-order optimization, to stabilize the search process of DARTS. So, I keep my rating as 6.

---

### Official Review · Reviewer_JSTM · 2021-11-03

**Correctness:** 3
**Technical Novelty And Significance:** 4
**Empirical Novelty And Significance:** 3
**Recommendation:** 6
**Confidence:** 4

**Main Review:**

The instability of DARTS has always been an topic in the NAS area, and has drawn a lot of attention. Many works have investigated the issue and propose methods to alleviate it.

This paper, conducts analysis on the first/second order approximation of network weights and points out that such approximation leads to the instable optimization of architecture parameters. Consequently it proposes to solve the problem via zero-order optimization, and propose three zero-order optimization methods, which is very novel. Experiments demonstrate the effectiveness of the proposed methods.

The weakness:
1. The increased search cost due to the sampling and estimation process.
2. Though the experimental results improve against original DARTS, however these tasks are explored by plenty of works and an error rate of around 2.50 and 24.3 on CIFAR-10 and ImageNet are achieved by many works and are not that outstanding. More experiments may strengthen the results.

Some questions for the authors:
1. In ZARTS-GLD, it seems that the network weight $w^{*}$ is estimated by the current weight $w$ and the method searches within a small area around current architecture parameter $\alpha$. Is this estimation accurate and why?
2. In section 5.1, convergence analysis, do the architectures derived every 25 epochs change or do they keep the same after several epochs?

Overall, I think this is a good work.

**Summary Of The Paper:**

This paper presents ZARTS, a zero-order optimization method for DARTS, to search without enforcing the approximation of the network weights. It conducts in-depth analysis on the first/second order approximation in DARTS, and points out that such approximation leads to bias and instability. Then the work proposes three zero-order optimization methods to solve the issue.

**Summary Of The Review:**

This work presents zero-order optimization methods for DARTS, which is very novel in the area. The paper is well-organized, with sufficient analysis and sound technical contribution. I think it is a good work.

---

> ### Author Response · Authors · 2021-11-19
> **Thank you for your feedback**
>
> Thanks for your nice comments. Please see our general response and the detailed reply below, which we hope will resolve your concern.
>
> >**Q1:** Correctness of ZARTS-GLD.
>
> **A1:** We hope our response can help to address your confusion and possible misunderstanding.
> In all three variants of ZARTS, the operation weights $\omega^*$ are estimated by performing gradient descent for $M=10$ iterations, including ZARTS-GLD. Specifically, we sample from a pretty large range (under the default hyper-parameters, there are 11 different unit spheres, larger than the predefined sampling number). Thus, ZARTS-GLD takes the next step based on rich observations, whose effectiveness is supported by our empirical results in Table 2.
>
> >**Q2:** In Sec.5.1, how does the derived architecture change every 25 epochs?}
>
> **A2:** We plot the architectures of normal and reduction cells derived every 25 epochs in Fig. 11 and Fig. 12 (Appendix) in hopes of resolving your concern. The discovered architectures vary during the first 100 epochs and become stable after that, and the topology remains unchanged after 150 epochs.
>
> >**Q3:** The search cost is high.
>
> **A3:** Please refer to our general response (Computational Cost), which we hope will resolve your concern.
>
> >**Q4:** The improvements of accuracy on CIFAR and ImageNet are not significant.
>
> **A4:** Please refer to our general response (Performance Advantage), which we hope will resolve your concern.

---

### Official Review · Reviewer_rwzN · 2021-11-05

**Correctness:** 3
**Technical Novelty And Significance:** 4
**Empirical Novelty And Significance:** Not applicable
**Recommendation:** 6
**Confidence:** 5

**Main Review:**

The empirical analyses are reasonable and sound. The proposal to use zero-order optimization methods is novel and interesting. The paper is well-written. I mainly have the following concerns.

1.	Since the authors use zero-order methods to optimize the architecture parameters, there is no requirement that the loss is differentiable wrt the architecture parameters. In this case, can we optimize the binary architecture parameters directly without the need to introduce the continuous softmax relaxation?
2.	The zero-order optimization methods introduce much more computational cost than the first-order gradient-based method, which causes the high search cost. So, it would be difficult to directly search on ImageNet. Considering many NAS methods have already been able to directly search on ImageNet, this drawback limits the potential applicability of the proposed methods.
3.	I think the stability of a NAS method can be claimed only when the method is able to produce similar searc results stably. Can the authors prove that multiple implementations of the proposed search method would lead to the same architecture or similar architectures with close performances? The result of Figure 2 is interesting. But I think the trends in Figure 2 are of randomness and difficult to reproduce. It is better to report the averaged trends of multiple implementations.
4.	The search results are not significantly better than some baselines. On ImageNet, the result is not better than PC-DARTS (ImageNet) that has 24.2% err. as reported in [1]. Besides, many studies are not cited and compared with, such as [2,3,4,5].
5.	Is the proposed method applicable to the chain-based search space adopted in [6,7,8]?


[1] Xu et al., PC-DARTS: PARTIAL CHANNEL CONNECTIONS FOR MEMORY-EFFICIENT ARCHITECTURE SEARCH

[2] Liang et al., DARTS+: Improved Differentiable Architecture Search with Early Stopping

[3] Chu et al., DARTS-: ROBUSTLY STEPPING OUT OF PERFORMANCE COLLAPSE WITHOUT INDICATORS

[4] Yang et al., ISTA-NAS: Efficient and Consistent Neural Architecture Search by Sparse Coding

[5] Zhou et al., Theory-Inspired Path-Regularized Differential Network Architecture Search

[6] Chu et al., FairNAS: Rethinking Evaluation Fairness of Weight Sharing Neural Architecture Search

[7] Yu et al., BigNAS: Scaling up Neural Architecture Search with Big Single-stage Models

[8] Mei et al., AtomNAS: FineGrained End-to-End Neural Architecture Search


**Summary Of The Paper:**

This study proposes to use zero-order optimization methods for neural architecture search based on the empirical observation that the approximation of differentiability will distort the loss landscape and lead to the biased objective and inaccurate gradient. Experiments on multiple datasets are conducted to show the improved stability and performance of the search results.

**Summary Of The Review:**

I vote for a weak accept currently considering the novelty of the proposed methods. But the concerns above need to be well addressed.

---

> ### Author Response · Authors · 2021-11-19
> **Thank you for your feedback [1/2]**
>
> Thank you for the time and constructive feedback. Please see our general response and the following detailed reply, which we hope will resolve your concern.
>
> >**Q1:** Can ZARTS optimize the binary architecture parameters directly without the continuous softmax relaxation?
>
> **A1:** Thanks for the interesting suggestion. However, directly optimizing binary architecture parameters belongs to discrete optimization, which has not been supported by our method.
> In contrast, the zero-order optimization methods, including RS, MGS, and GLD adopted in our ZARTS, aim to solve the continuous optimization problem in which gradients w.r.t. the variables are intractable or hard to obtain. As a possible combination, we can convert the discrete optimization to a continuous one by introducing un-differentiable but more effective regularization terms, such as L0 and L1 norm, to restrict the number of non-zero items in architecture parameters, which we are willing to explore in our future work.
>
> >**Q2:** High search cost limits the potential applicability of ZARTS.
>
> **A2:** Please refer to our general response (Computational Cost), which we hope will resolve your concern.
>
> >**Q3:** Can ZARTS stably discover the same architecture or architectures with close performances? And how is the stability of ZARTS?
>
> **A3:** First, we would like to clarify the "stability" discussed in our work (Sec. 5.1). Specifically, it refers to the ability to overcome the instability issue of DARTS [1,2] that parameterless operations (especially identity operation) tend to dominate the normal cell during the search process.
> R-DARTS reveals that the instability issue occurs when searching on four specially-designed search spaces. Amended-DARTS shows that a long period of searching, such as 200 epochs, will also aggravate the instability issue.
> Therefore, we verify the stability of our method by the above two metrics: performance on the four spaces (as reported in Table 2) and trend of performance along with searching of 200 epochs (as illustrated in Fig. 2).
>
> Moreover, we add new experimental results and analysis to the Appendix (Fig. 5 and Sec. B.5) to supplement Fig. 2. Specifically, Recent works [1,2,3,4] show that too many identity operations (more than 4) are typical of the instability issue. Therefore, we utilize the number of parameterless operations and identity operations in one normal cell to represent the performance of architectures since it is time-consuming to train every derived architecture along the search process.
> Fig. 5 illustrates the number of identity operations of our ZARTS along with three peer methods, including DARTS, PC-DARTS and S-DARTS.
> We observe that the architectures discovered by DARTS, PC-DARTS, and S-DARTS will be gradually dominated by parameterless operations (especially identity operations), implying that the instability issue occurs. In contrast, our ZARTS can stably control the number of parameterless operations.
>
> Finally, we want to claim that ZARTS can stably discover exemplary architectures with consistently high performance, as verified by the average performance and standard deviation of parallel tests shown in Table 3.
> Specifically, we conduct four parallel tests by searching under different random seeds, and our ZARTS achieves 2.54\%$\pm$0.07 average performance on CIFAR-10 and 16.29\%$\pm$0.53 on CIFAR-100, outperforming other prior methods.
>
> [1] Zela et al., Understanding and robustifying differentiable architecture search. in ICLR, 2020.
>
> [2] Bi et al., Stabilizing darts with amended gradient estimation on architectural parameters.
>
> [3] Chen et al., Progressive Differentiable Architecture Search: Bridging the Depth Gap between Search and Evaluation. In ICCV, 2019.
>
> [4] Chen et al., Stabilizing differentiable architecture search via perturbation-based regularization. In ICML, 2020.
>
> >**Q4:** Missing comparison with some related works and the results of ZARTS are not significantly better than some baselines (PC-DARTS achieves 24.2\% on ImageNet).
>
> **A4:** Thanks for sharing with us these related works, and we add them to Table 3 in our new version. Our method slightly outperforms ISTA-NAS and DARTS- on CIFAR-10 and outperforms PR-DARTS and DARTS- on CIFAR-100 by 1\% on CIFAR-100 dataset.
> As for the comparison with PC-DARTS, the quoted result in our paper is its transferred performance for fair comparison (search on CIFAR-10 and transfer to ImageNet). Specifically, PC-DARTS achieves 74.9\% accuracy, and our ZARTS achieves 75.7\% on ImageNet.
>
> As for the performance advantage over peer methods, please refer to the general response (Performance Advantage).

---

> > ### Author Response · Authors · 2021-11-19
> > **Thank you for your feedback [2/2]**
> >
> > > **Q5:** Is the proposed method applicable to a large chain-based search space?
> >
> > **A5:** Our method belongs to a one-shot based NAS and is independent of the search space. As long as a supernet (one-shot model) is built, ZARTS can be adopted.
> > However, it is hard to train supernets for a large chain-based search space due to the limitation of GPU memory, which is a common shortcoming of one-shot based methods, including our ZARTS, DARTS, FaireDARTS and R-DARTS.
> > Single-path based methods such as GDAS and ROME can be a cure for this issue, which we leave as our future work as discussed in Sec. 6.

---

### Author Response · Authors · 2021-11-19
**General Response by Authors**

Dear area chair and reviewers,

We appreciate the reviewers' time and valuable comments. Overall, the reviewers thought our paper was well-written (Reviewer rwzN and JSTM) and acknowledged our novelty (Reviewer rwzN and JSTM), theoretical soundness (Reviewer rwzN, JSTM, and 5Kkb), sufficient analysis (Reviewer rwzN, JSTM, and 5Kkb), and comprehensive evaluations (Reviewer rwzN, JSTM, and 5Kkb). The primary concerns lie in the search cost and technical novelty (Reviewer 5Kkb) of our method. However, we believe there might exist a misunderstanding regarding our motivation, which led the reviewers to underestimate the novelty of our method. Here we first restate our motivation and contributions in order to resolve some big picture issues.

#### **Motivation \& Contributions**
In this work, we rethink the bi-level optimization for NAS and empirically show that inaccurate estimation (first/second-order approximation) for optimal network weights ${\omega}^*({\alpha})$ distorts the search loss landscape and shifts the global minimum (Fig. 3), leading to the instability issue (Sec. 3.2). However, DARTS has to adopt such sketchy approximation in order to compute the gradient of loss function w.r.t. ${\alpha}$. This work fundamentally circumvents the use of gradient descent (first-order optimization) and turns to zero-order optimization, thus allowing for accurate estimation for ${\omega}^*({\alpha})$. Extensive analysis and experiments support that zero-order optimization algorithms can be an effective cure for the instability problem (Sec. 5.1).

We believe this paper provides a pioneering perspective of applying zero-order optimization to the NAS area and proves its advantage in alleviating the instability issue. Specifically, we propose a general framework based on sampling (Alg. 1), which is compatible with various algorithms and investigate three variants in depth. The theoretical part of this paper bridges first- and zero-order optimization regarding the NAS problem and analyzes the relationship between ZARTS and DARTS (Sec. 4.4). We show that ZARTS-RS/-MGS can be seen as an expansion of DARTS, and they degrade to DARTS in the condition of first/second-order approximation.

Finally, we want to argue that designing a brand-new zero-order optimization algorithm for general cases is complicated and out of the scope of this paper, which we are willing to explore in the future.

#### **Computational Cost**
We agree that the search cost of ZARTS is higher than some recent works aiming to speed up the search process. However, this work focuses on revealing the neglected harm of inaccurate approximation for optimal network weights ${\omega^*({\alpha})}$ in DARTS. To this end, we propose an efficient search method ZARTS that can robustly discover architectures in 1.0 GPU-day, which is more efficient than many other methods that aim to stabilize DARTS, such as RDARTS, S-DARTS and Amended-DARTS (as compared in Table 3).

Moreover, we argue that the prior works aiming to speed up the search process, such as PC-DARTS and GDAS, are orthogonal to our method. In fact, the fast search speed of PR-DARTS also results from the operation sampling strategy, similar to ProxylessNAS and GDAS. Therefore, we believe those techniques could potentially be combined with our method to see further benefits, as discussed in Sec. 6.

#### **Performance Advantage**
R-DARTS suggests comparing the average performance of multiple parallel tests to evaluate NAS methods. As shown in Table 2, the average performance of our ZARTS outperforms others by more than 1\% on CIFAR-100 dataset.

Additionally, we would like to highlight our solution to the instability issue of DARTS, which is analyzed in Sec. 5.1 and shown in Table 2 and Fig. 2. We also supplement new experimental results in Sec. B.5 and Fig. 5 (Appendix) to further strengthen the effectiveness of our method.
Though many prior methods achieve competitive performance, they usually involve human knowledge of architecture. For example, P-DARTS restricts the number of skip-connection as two, DARTS- introduces auxiliary skip-connection to suppress $\alpha_{skip}$, and DARTS+ adopts early stop strategy before skip-connection dominate the cell.
On the contrary, this work explores the impact of first-order approximation on the search performance, which is neglected by other works, and introduces an efficient and robust search method to alleviate the instability issue of DARTS by zero-order optimization algorithm, whose effectiveness and stability is verified in Table 2 and Fig. 2.

Moreover, this work refines the bi-level optimization for the one-shot NAS method and is orthogonal to those exploring better discretization processes such as SGAS and DARTS-PT, which we believe can be combined with ZARTS to further improve the performance.

In the individual responses, we provide detailed answers to all the questions point-by-point, which we hope will resolve reviewers' concerns.

---

> ### Author Response · Authors · 2021-11-19
> **Summary of Updates**
>
> We thank all reviewers for their detailed and thorough comments. We have added some new content to our paper, and here is a summary of updates.
>
> - Comparison of ZARTS with ISTA-NAS, DARTS- and PR-DARTS is added to Table 3 (suggested by Reviewer rwzN). ZARTS achieves state-of-the-art on CIFAR-100 and outperforms ISTA-NAS and DARTS- on CIFAR-10.
> - Expanded discussion on convergence ability of three implementations of ZARTS is added to  Sec. B.5 and Fig. 5 in Appendix (proposed by Reviewer rwzN). ZARTS has the best convergence ability compared to DARTS, PC-DARTS, and S-DARTS.
> - Normal and reduction cells derived every 25 epochs during the search process of 200 epochs are illustrated in Fig. 11 and Fig. 12 in Appendix (proposed by Reviewer JSTM). The discovered architecture becomes stable after 150 epochs.
> - Results of three implementations of ZARTS on the search space of S-DARTS are added to Table 7 in Appendix (proposed by Reviewer 5Kkb). ZARTS achieves state-of-the-art on 7 benchmarks.

---

### Decision · Program_Chairs · 2022-01-20

**Decision:**

Reject

**Comment:**

The paper proposes a series of zeroth order optimization approaches to stabilize DARTS training. Although the reviewers think that zeroth order approach is novel to the NAS community, they also point out several weaknesses. In particular, the method will introduce extra computation time and the results are not really standing out comparing with other state-of-the-art methods. Therefore, despite some interesting ideas are presented in the paper, we decide to reject the paper and encourage the authors to address those weaknesses in their future revision.